# EHRCon: Dataset for Checking Consistency between Unstructured Notes and Structured Tables in Electronic Health Records

**Yeonsu Kwon**[1][*], **Jiho Kim**[1][*], **Gyubok Lee**[1], **Seongsu Bae**[1], **Daeun Kyung**[1],
**Wonchul Cha**[2], **Tom Pollard**[3], **Alistair Johnson**[4], **Edward Choi**[1]
[1]KAIST  [2]Samsung Medical Center  [3]MIT  [4]University of Toronto
{yeonsu.k, jiho.kim, edwardchoi}@kaist.ac.kr

## Abstract

Electronic Health Records (EHRs) are integral for storing comprehensive patient medical records, combining structured data (*e.g.*, medications) with detailed clinical notes (*e.g.*, physician notes). These elements are essential for straightforward data retrieval and provide deep, contextual insights into patient care. However, they often suffer from discrepancies due to unintuitive EHR system designs and human errors, posing serious risks to patient safety. To address this, we developed `EHRCon`, a new dataset and task specifically designed to ensure data consistency between structured tables and unstructured notes in EHRs. `EHRCon` was crafted in collaboration with healthcare professionals using the MIMIC-III EHR dataset, and includes manual annotations of 4,101 entities across 105 clinical notes checked against database entries for consistency. `EHRCon` has two versions, one using the original MIMIC-III schema, and another using the OMOP CDM schema, in order to increase its applicability and generalizability. Furthermore, leveraging the capabilities of large language models, we introduce CheckEHR, a novel framework for verifying the consistency between clinical notes and database tables. CheckEHR utilizes an eight-stage process and shows promising results in both few-shot and zero-shot settings. The code is available at `https://github.com/dustn1259/EHRCon`.

## 1 Introduction

Electronic Health Records (EHRs) are digital datasets comprising the rich information of a patient's medical history within hospitals. These records integrate both structured data (*e.g.*, medications, diagnoses) and detailed clinical notes (*e.g.*, physician notes). The structured data facilitates straightforward retrieval and analysis of essential information, while clinical notes provide in-depth, contextual insights into the patient's condition. These two forms of data are interconnected and provide complementary information throughout the diagnostic and treatment processes. For example, a practitioner might start by reviewing test results stored in the database, then determine a diagnosis and formulate a treatment plan, which are documented in the clinical notes. These notes are subsequently used to update the structured data in the database.

However, inconsistencies can arise between the two sets of data for several reasons. One primary issue is that EHR interfaces are often designed with a focus on administrative and financial tasks, which makes it difficult to accurately document clinical information [32]. Additionally, overburdened practitioners might unintentionally introduce errors by importing incorrect medication lists, copying and pasting outdated records, or entering inaccurate test results [4, 24, 38]. These errors can lead

---

[*]These authors contributed equally

38th Conference on Neural Information Processing Systems (NeurIPS 2024) Track on Datasets and Benchmarks.

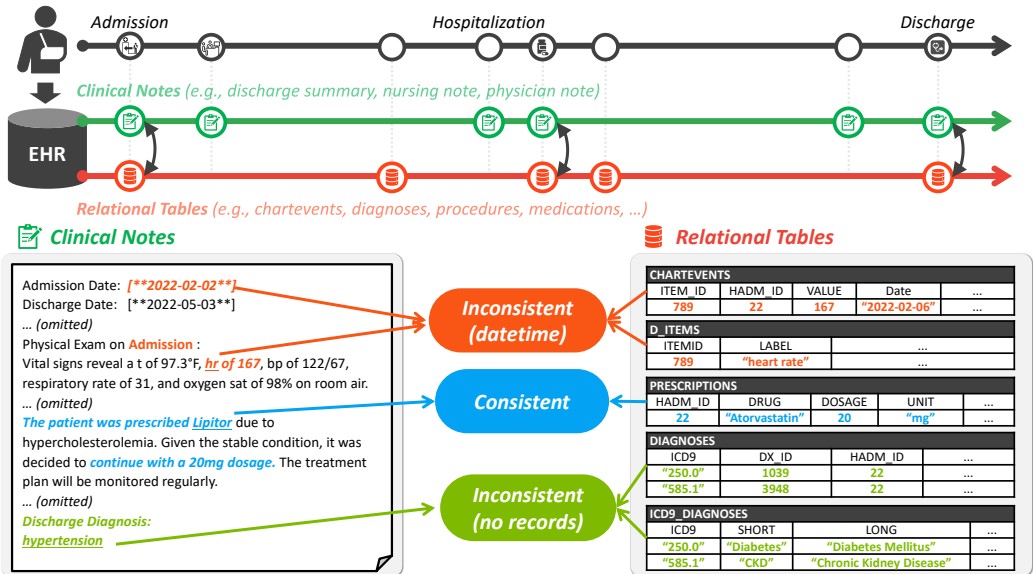

Figure 1: Examples of consistent and inconsistent data between clinical notes and EHR tables: An inconsistent example (datetime) is when a clinical note records an HR (abbreviation for heart rate) of 167 on "2022-02-02" but the EHR table shows the same HR on "2022-02-06". A consistent example is when both the clinical note and the EHR table document the administration of Atorvastatin with matching drug name, dosage, and unit. Another example of inconsistency occurs when a clinical note mentions a hypertension diagnosis, but the EHR table lacks this information.

to significant discrepancies between the structured data and clinical notes in the EHR, potentially jeopardizing patient safety and leading to legal complications [3].

Manual scrutiny of these records is both time-intensive and costly, underscoring the necessity for automated interventions. Despite the need for automated systems, previous studies on consistency check between tables and text have primarily focused on single claims and small-scale single tables [1, 7, 8, 35]. These approaches are not designed for the complex and large-scale nature of EHRs, which require more comprehensive and scalable solutions.

To this end, we propose a new task and dataset called EHRCon, which is designed to verify the consistency between clinical notes and large-scale relational databases in EHRs. We collaborated closely with practitioners[2] to design labeling instructions based on their insights and expertise, authentically reflecting real hospital environments. Based on these labeling instructions, trained human annotators used the MIMIC-III EHR dataset [15] to manually compare 4,101 entities mentioned in 105 clinical notes against corresponding table contents, annotating them for CONSISTENT or INCONSISTENT as illustrated in Figure 1. Our dataset also offers interpretability by including detailed information about the specific tables and columns where inconsistencies occurred. Moreover, it contains two versions, one based on the original MIMIC-III schema, and another based on its OMOP CDM [34] implementation, allowing us to incorporate various schema types and enhance the generalizability.

Additionally, we introduce CheckEHR, a framework that leverages the reasoning capabilities of large language models (LLMs) to verify consistency between clinical notes and tables in EHRs. CheckEHR comprises eight sequential stages, enabling it to address complex tasks in both few-shot and zero-shot settings. Experimental results indicate that in a few-shot setting, our framework achieves a recall performance of 61.06% on MIMIC-III and 54.36% on OMOP. In a zero-shot setting, it achieves a recall performance of 52.39% on MIMIC-III. Additionally, we conduct comprehensive ablation studies to thoroughly analyze the contributions of each component within CheckEHR.

---

[2]EHR technician, nurse, and emergency medicine specialist with over 15 years of experience

## 2 Related Works

**Consistency Check**  Fact verification involves assessing the truthfulness of claims by comparing them to evidence [13, 18, 23, 26–31, 33]. This task is similar to ours as it involves checking for consistency between two sets of data. Among the various datasets, those utilizing tables as evidence are particularly relevant. TabFact [7], a prominent dataset for table-based fact verification, focuses on verifying claims by reasoning with Wikipedia tables. Additionally, INFOTABS [8] uses info-boxes from Wikipedia, and SEM-TAB-FACTS [35] utilizes tables from scientific articles. Furthermore, FEVEROUS [1] is a dataset designed to verify claims by reasoning over both text and tables from Wikipedia. While these datasets focus on verifying individual claims with small-scale tables (*e.g.*, most 50 rows), our methodology differs significantly. We handle entire clinical notes where multiple claims must be first recognized, then perform consistency checks against a larger heterogeneous relational database (*i.e.*, 13 tables each with up to 330M rows). This requires a more comprehensive and scalable solution for fact verification. Consequently, our work extends beyond previous studies, presenting a novel task in the field of Natural Language Processing (NLP) as well as healthcare.

**Compositional Reasoning**  Large Language Models (LLMs) [2, 6, 20, 21] have demonstrated remarkable abilities in handling a wide range of tasks with just a few examples in the prompts (*i.e.*, in-context learning). However, some complex tasks remain challenging when tackled through in-context learning alone. To address these challenges, researchers have developed methods to break down complex problems into smaller, more manageable sub-tasks [16, 19, 39]. These decomposition techniques have also been applied to tasks that involve reasoning from structured data [12, 17, 36]. One significant development is StructGPT [12], which enables LLMs to gather evidence and reason using structured data to answer questions. Inspired by these decomposition techniques, CheckEHR improves the accuracy and efficiency of consistency checks between clinical notes and tables, effectively overcoming the limitations of in-context learning methods.

## 3 EHRCon

EHRCon includes annotations for 4,101 entities extracted from 105 randomly selected clinical notes, evaluated against 13 tables within the MIMIC-III database [15].[3] MIMIC-III contains data from approximately 40,000 ICU patients treated at Beth Israel Deaconess Medical Center between 2001 and 2012, encompassing both structured information and textual records. To enhance standardization, we also utilize the Observational Medical Outcomes Partnership (OMOP) Common Data Model (CDM)[4] version of MIMIC-III. OMOP CDM, a publicly developed data standard designed to unify the format and content of observational data for efficient and reliable biomedical research. In this regard, developing the OMOP version of EHRCon will be highly beneficial for future research scalability. In this section, we detail the process of designing the labeling instructions (Sec. 3.1), and labeling the dataset (Sec. 3.3) on MIMIC-III dataset. The labeling for the OMOP CDM version and detailed data preparation steps are provided in Appendix B.

### 3.1 Labeling Instructions

To reflect actual hospital environments, practitioners and AI researchers collaboratively designed the labeling instructions. The following are the three important aspects of the labeling instruction, and more detailed instructions can be found in Appendix C.

**Labels**  We classify the entities as either CONSISTENT or INCONSISTENT based on their alignment with the tables. This approach is fundamentally different from traditional fact-checking methods, which typically determine whether claims are SUPPORTED or REFUTED using texts or tables as definitive evidence. In contrast, in the context of EHR, both tables and clinical notes can contain errors, making it impossible to define one as definitive evidence. Therefore, a more flexible approach is used by labeling them as CONSISTENT or INCONSISTENT. An entity is labeled as CONSISTENT if all related information, such as values and dates in the note, matches exactly with the tables. Conversely, if even one value differs, it is labeled as INCONSISTENT.

---

[3]Although MIMIC-IV [14] is more recent than MIMIC-III, we use MIMIC-III in this work because MIMIC-IV lacks diverse note types (such as physician notes and nursing notes), and is missing all dates in notes for de-identification, as opposed to shifting the dates in MIMIC-III notes.

[4]https://www.ohdsi.org/data-standardization/

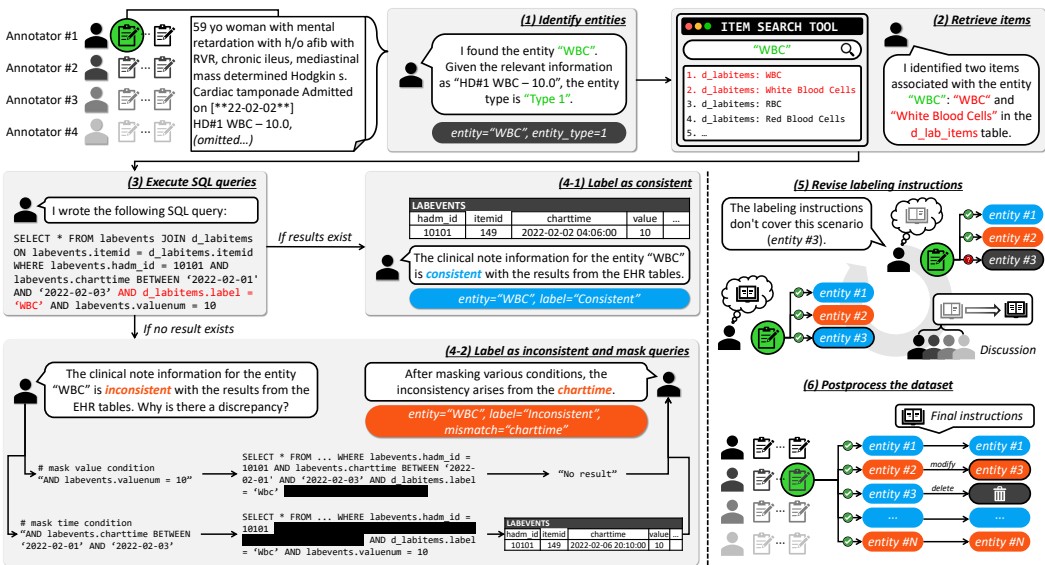

Figure 2: Annotation process of `EHRCon`: The annotation process involves annotators reviewing clinical notes, identifying and classifying entities into Type 1 and Type 2, and extracting relevant information to generate and execute SQL queries. If the SQL queries yield no results, conditions (*e.g.*, value or time) are masked to pinpoint where the inconsistency occurred. When annotators encounter corner cases, they update the labeling instructions through discussion. After all labeling is complete, a post-processing phase is conducted to ensure high-quality data.

**Definition of Entity Types**  We categorized the entities in the notes into two main types for labeling. First, entities with numerical values, such as "*WBC 10.0*", are defined as Type 1. Second, entities without values but whose existence can be verified in the database, such as "*Vancomycin was started.*", are labeled as Type 2. In our study, we did not label entities with string values because they can be represented in various ways within a database. For example, the phrase "*BP was stable.*" might be shown in a *value* column as "*Stable*" or "*Normal*", or it might be indicated by numeric values in the database. This variability can lead to labeling errors. However, to support future research, we included them as Type 3 entities in our dataset, but did not use them in the main experiments.

**Time Expression**  Clinical notes contain various time expressions, so we manually analyzed the time expressions in these notes (see Appendix D). As a result, we found that they can be categorized into three groups: 1) event time written in a standard time format, 2) event time described in a narrative style, and 3) time information of the event not written. When an entity is presented in the standard date and time format (*i.e.*, *YYYY-MM-DD*, *YYYY-MM-DD HH:MM:SS*), we validate whether a clinical event occurred exactly at that timestamp. For narrative-style expressions, such as "*around the time of patient admission*" or "*shortly after discharge*", we consider records within the day before and after the specified date to account for the approximate nature of the timing. In cases where no precise time information is provided, we determine the relevant time frame based on the type of the note. For instance, in discharge summaries, we examine the entire admission period, while for physician and nursing notes, we check the records within one day before and after the chart date.

## 3.2 Item Search Tool

Clinical notes can include a mix of abbreviations (*e.g.*, *Temp* vs. *Temperature*), common names (*e.g.*, *White Count* vs. *White Blood Cells*), and brand names (*e.g.*, *Tylenol* vs. *Acetaminophen*) depending on the context and the practitioner's preference. This discrepancy causes issues where the entities noted in the clinical notes do not match exactly with the items in the database. To resolve this, we developed a tool to search for database items related to the note entities.

To create a set $E$ of database items related to the entity $e$, we followed a detailed approach. First, we used the C4-WSRS medical abbreviation dataset [25] to gather a thorough list of abbreviation-full

Table 1: Data statistics of `EHRCon`.

| Note Type | Entity | | | Labels | Note | |
|---|---|---|---|---|---|---|
| | Mean Num | Total Num | Type 1 / 2 | Con. / Incon. | Total Num | Mean Length |
| Discharge Summary | 50.21 | 1,908 | 1,400 / 508 | 1,181 / 727 | 38 | 2,789 |
| Physician Note | 46.36 | 1,530 | 1,111 / 419 | 1,230 / 300 | 33 | 1,859 |
| Nursing Note | 19.50 | 663 | 500 / 163 | 522 / 141 | 34 | 1,111 |
| Total | 39.06 | 4,101 | 3,011 / 1,090 | 2,933 / 1,168 | 105 | 1,953 |

name pairs. Then, we utilized GPT-4 (0613) [20][5] to extract medication brand names from clinical notes and convert them to their generic names. By combining these methods, we built an extensive set $V$, which includes abbreviations, full names, brand names, and generic names associated with the entity $e$. Finally, to create the set $E$, we calculated the bi-gram cosine similarity scores between the elements in $V$ and the items in our database, retrieving those that exceeded a specific threshold.

### 3.3 Annotation Process

In this section, we explain the data annotation process depicted in Figure 2. Annotators begin by carefully reviewing the clinical notes, utilizing web searches and discussions with GPT-4 (0613). Through this process, they identify entities and relevant information within the notes. Subsequently, the identified entities are classified into Type 1 and Type 2, as outlined in Sec. 3.1 (Figure 2-(1)). For each entity, annotators use the Item Search Tool (Sec. 3.2) to find the relevant items in the database (Figure 2-(2)). They then select the items and tables associated with the entity. If none of the retrieved items match the entity, the annotators manually find and match the appropriate items. Following this, the annotators extract information related to the entity from the notes (*e.g.*, dates, values, units) and use them to generate SQL queries, as explained in Appendix P (Figure 2-(3)). Finally, the annotators execute the generated queries and review the results to label the entity as either CONSISTENT or INCONSISTENT (Figure 2-(4)). If a query yields no results, the SQL conditions are sequentially masked and executed to pinpoint the source of the inconsistency (Figure 2-(4)-2). Also, when the annotators encounter a corner case that is not addressed in the existing instructions, they update the instructions after thorough discussion (Figure 2-(5)). Upon completing all annotations, the annotators engaged in a post-processing phase to ensure high-quality data. This phase involved additional annotation of entities according to the final labeling instructions, as well as the removal of any misaligned entities (Figure 2-(6)). We implemented additional quality control processes to ensure high-quality. For more details on these processes, see Appendix E.

### 3.4 Statistics

**Inconsistencies Found in Notes** As seen in Table 1, discharge summaries account for a significant portion of inconsistent cases, with 727 out of 1,168 total cases. Unlike nursing and physician notes, which document clinical events as they occur, discharge summaries are written at the time of discharge and summarize major events and treatments. This timing could potentially increase the likelihood of errors. Given the pivotal role discharge summaries play in hospitals such as during inpatient-outpatient transitions [5], inconsistencies in these notes can negatively impact patient care.

**Inconsistencies Found in Tables** We found 36.55% of inconsistencies in the labevents table and 17.58% in medication-related tables (*e.g.*, prescriptions). These data are crucial for patient care, and such discrepancies can lead to misdiagnosis and inaccurate medication administration, potentially resulting in patient death [9, 22]. Therefore, implementing automated consistency checks is important to ensure data accuracy and consistency.

**Inconsistencies Found in Columns** An in-depth analysis revealed that 56.16% of discrepancies are related to time, with 58.23% of these temporal inconsistencies involving a one-hour difference between tables and clinical notes, possibly due to issues in the EHR system [37]. This suggests that the discrepancy could result from not only human but also software issues. For a more detailed analysis, refer to Appendix F.

---

[5]In all cases where GPTs were used, the HIPAA-compliant GPT models provided by Azure were used.

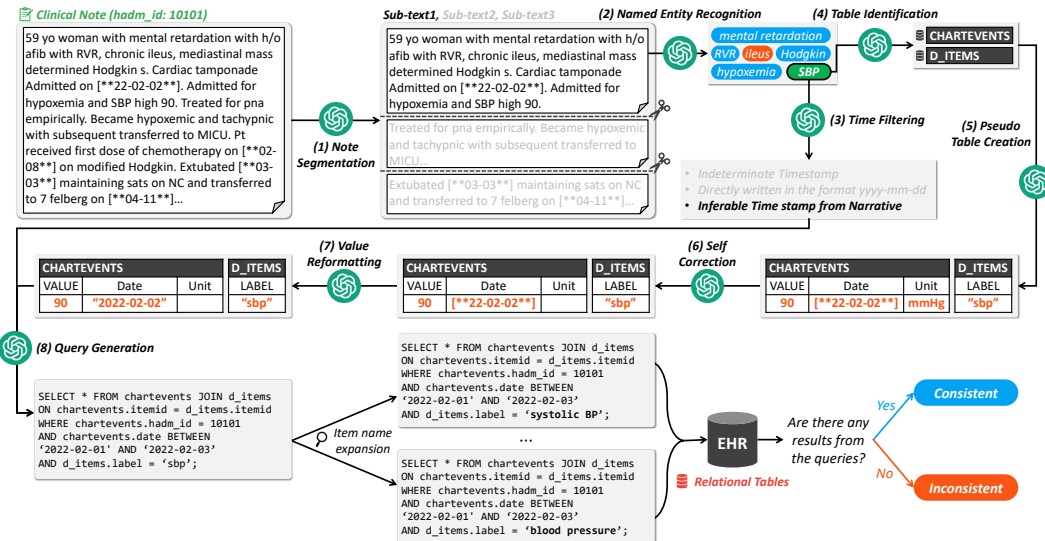

Figure 3: Overview of CheckEHR. The framework consists of eight distinct stages: Note Segmentation, Named Entity Recognition, Time Filtering, Table Identification, Pseudo Table Creation, Self-Correction, Value Reformatting, and Query Generation.

# 4 CheckEHR

CheckEHR is a novel framework designed to automatically verify the consistency between clinical notes and a relational database. As depicted in Figure 3, CheckEHR encompasses eight sequential stages: Note Segmentation, Named Entity Recognition (NER), Time Filtering, Table Identification, Pseudo Table Creation, Self Correction, Value Reformatting, and Query Generation. All stages utilize the in-context learning method with a few examples to maximize the reasoning ability of large language models (LLMs). The prompts for each step are included in Appendix G.

**Note Segmentation** LLMs face significant challenges in processing long clinical notes due to its limitations in handling extensive context lengths. To overcome this challenge, we propose a new scalable method called Note Segmentation, which divides the entire clinical note into smaller sub-texts that each focus on a specific topic. The following outlines the process of creating a set $\mathcal{T}$, composed of sub-texts from clinical note $P$. First, the text $P$ is divided into two parts: $P_0^f$, containing the first $l$ tokens, and the remaining text, $P_0^b$. Then, $P_0^f$ is segmented by the LLM into $n$ sub-texts, each with its own distinct topic: $\{P_{0,1}^f, P_{0,2}^f, ..., P_{0,n}^f\}$.[6] The sub-texts from $P_{0,1}^f$ to $P_{0,n-1}^f$ are added to the set $\mathcal{T}$. Since $P_{0,n}^f$ is likely incomplete due to the $l$ token limit, it is concatenated with $P_0^b$ for further segmentation. The combined text of $P_{0,n}^f$ and $P_0^b$ is referred to as $P_1$. This segmentation continues until the length of $P_i$ is $l$ tokens or less, at which point $P_i$ is added to $\mathcal{T}$. To ensure smooth transitions, each sub-text includes some content from adjacent sub-texts. The algorithm and conceptual figure of Note Segmentation are detailed in Appendix H.

**Named Entity Recognition** Our task takes the entire text as input, making the extraction of named entities essential for consistency checks. In this task, the LLM extracts entities related to the 13 tables, focusing on those with clear numeric values, and those whose existence can be verified in the database even without explicit values.[7] This selective extraction is crucial for maintaining the accuracy and reliability of our checks.

---

[6]$l$ is determined by the context length of the LLM. In this study, $l$ was set to 1000 and $n$ to 3.

[7]Narrowing down the NER target like this might seem like taking advantage of our knowledge from the dataset construction process. However, we would like to emphasize that the scope of named entities is part of the task definition, and it is essential to share this information with the model so that the model at least understands the objective.

**Time Filtering**   At this stage, the LLM determines whether the time expression of a clinical event is in a specific time format, written in a narrative style, or if the time is not specified. The results from this step are utilized for generating queries at the last stage.

**Table Identification**   To create a pseudo table in the next stage, it is essential to identify the relevant tables related to the entities. At this stage, the LLM uses table descriptions, foreign key relationships, and just two example rows to identify the necessary table names.

**Pseudo Table Creation**   Since clinical notes include content that cannot be easily verified through tables, the LLM creates a pseudo table to effectively extract table-related information. The LLM extracts the information through a multi-step process as follows: First, extracts sentences from the clinical note that contain the entity to verify. Then, analyzes the extracted sentences to determine the time information of the entity. Finally, completes the pseudo table by extracting information about the remaining columns (*e.g.*, value, unit) from the notes, using the previously obtained information. Examples of the detailed process for creating the pseudo table can be found in Appendix I.

**Self Correction**   During the construction of the pseudo table, we found that hallucinations by the LLM were frequent (see Appendix J). For example, there were instances where the LLM generated unit information that was not present in the notes. To address this issue, the LLM re-evaluates whether the pseudo table created in the previous stage is directly aligned with the notes. We then use only the results that are actually aligned.

**Value Reformatting**   In clinical notes and tables, the same information may be expressed differently. For instance, a clinical note might mention '*admission*', while the table might record the admission date as '*2022-02-02*'. To align the data types between the generated pseudo table and the actual table, the LLM reformats the pseudo table by using the schema information.

**Query Generation**   Using the results from the Time Filtering and Value Reformatting, the LLM creates an SQL query. This query is executed against the database to check if the content mentioned in the notes matches the actual database content. During execution, we replace the entity in the SQL query with items retrieved from the database. This process involves leveraging the Item Search Tool (see Sec. 3.2) to cover both medication brand names and their corresponding abbreviations.

# 5   Experiments

## 5.1   Experimental Setting

**Base LLMs**   We aim to conduct an evaluation of our EHRCon and our proposed framework CheckEHR. For this evaluation, we utilized Tulu2 70B [10], Mixtral 8X7B [11], Llama-3 70B[8], and GPT-3.5 (0613) [21] as the base LLMs within our framework. To effectively measure CheckEHR's performance, experiments were conducted under both few-shot and zero-shot settings.

**Note Pre-processing**   We filtered out information from the notes that is difficult to confirm from the tables (*e.g.*, pre-admission history) to focus on the current admission records (see Appendix B.1.1). All experiments used these processed notes, with results using the unfiltered original notes available in Appendix K.

**Metrics**   We evaluate CheckEHR's performance for each note using Precision, Recall and Intersection, then calculate the average across all notes. Precision is the number of correctly classified entities divided by the number of all recognized entities[9] in the note. Recall is the number of correctly classified entities divided by the number of all human-labeled entities in the note. Intersection is the number of correctly classified entities divided by the number of *correctly* recognized entities in the note. Note that we use Intersection to assess how well CheckEHR performs at least for the correctly recognized entities, considering the difficulty of NER. Details on the experiment setups are in Appendix L.

---

[8]https://llama.meta.com/llama3

[9]Note that CheckEHR must first *recognize* entities in the notes during the NER stage before classifying them as CONSISTENT or INCONSISTENT.

Table 2: The main results of CheckEHR on MIMIC-III. Mixtral scored zero in the zero-shot setting, so it was not included in the table. Values in **bold** represent the highest performance for each metric among all models within the same shot setting.

| Shot | Models | Discharge Summary | | | Physician Note | | | Nursing Note | | | Total | | |
|---|---|---|---|---|---|---|---|---|---|---|---|---|---|
| | | Rec | Prec | Inters | Rec | Prec | Inters | Rec | Prec | Inters | Rec | Prec | Inters |
| Zero | Tulu2 | 11.82 | 27.48 | 46.92 | 9.1 | 20.15 | 40.83 | 15.32 | 23.23 | 30.37 | 12.08 | 23.62 | 38.37 |
| | Mixtral | - | - | - | - | - | - | - | - | - | - | - | - |
| | Llama-3 | 50.82 | 35.54 | 69.70 | 52.92 | 33.89 | 72.71 | 53.45 | 44.61 | 81.48 | **52.39** | 38.01 | **74.03** |
| | GPT-3.5 (0613) | 45.04 | 46.71 | 74.58 | 40.14 | 37.32 | 70.07 | 43.30 | 44.53 | 70.81 | 42.83 | **42.85** | 71.82 |
| Few | Tulu2 | 40.01 | 49.42 | 70.66 | 49.98 | 47.08 | 85.33 | 44.77 | 40.50 | 78.40 | 44.95 | 45.66 | 78.13 |
| | Mixtral | 54.70 | 49.76 | 71.21 | 53.71 | 37.97 | 83.48 | 69.86 | 49.65 | 85.01 | 54.70 | 45.79 | 79.90 |
| | Llama-3 | 50.44 | 47.01 | 76.25 | 56.11 | 42.75 | 84.30 | 52.60 | 38.08 | 75.96 | 53.05 | 42.61 | 78.83 |
| | GPT-3.5 (0613) | 64.31 | 54.64 | 81.60 | 54.64 | 44.01 | 81.41 | 64.25 | 47.25 | 95.74 | **61.06** | **48.63** | **86.25** |

Table 3: The main results of CheckEHR on MIMIC-OMOP. In this experiment, the NER step was skipped, and the gold entity was provided. The experiment was conducted in a few-shot setting. Values in **bold** represent the Total Recall and Total Precision from the model that performed better across the MIMIC-OMOP and MIMIC-III datasets.

| Data | Models | Discharge | | Physician | | Nursing | | Total | |
|---|---|---|---|---|---|---|---|---|---|
| | | Rec | Prec | Rec | Prec | Rec | Prec | Rec | Prec |
| **MIMIC-OMOP** | Tulu2 | 56.08 | 55.23 | 53.78 | 61.79 | 52.23 | 52.28 | **54.36** | 56.43 |
| | Mixtral | 53.20 | 54.86 | 48.50 | 63.90 | 58.16 | 59.29 | 53.28 | 59.35 |
| | Llama-3 | 53.83 | 58.22 | 53.82 | 76.95 | 59.49 | 60.06 | 55.71 | **65.07** |
| **MIMIC-III** | Tulu2 | 55.72 | 66.04 | 55.44 | 68.42 | 53.23 | 68.18 | 54.13 | **67.54** |
| | Mixtral | 55.23 | 65.20 | 63.55 | 57.78 | 69.62 | 68.25 | **62.80** | **63.74** |
| | Llama-3 | 53.85 | 63.28 | 68.19 | 64.45 | 64.25 | 63.35 | **62.09** | 63.69 |

## 5.2 Results

Table 2 presents the results of our framework for both the few-shot and zero-shot settings. Notably, using GPT-3.5 (0613) as the base LLM in the few-shot scenario achieves the best result, with a recall of 61.06%, precision of 48.63%, and an intersection score of 86.25%. This result demonstrates significantly improved performance compared to the direct use of GPT-3.5[10], which had a recall of 10.12% and a precision of 8.75%.

However, despite the carefully crafted 8-stage CheckEHR framework, the overall recall scores remain in the 40-60% range, underscoring the inherent difficulty of the task. This challenge is further underscored by the significant gaps between recall and intersection, and between precision and intersection. Such discrepancies indicate the difficulty of NER in our task. Despite providing the model all the entity extraction criteria defined for the task during the NER stage, both recall and precision performance remains low. This suggests that LLMs lack capabilities required to comprehend clinical notes and accurately extract only the entities that meet the criteria.

Furthermore, in our comparative analysis of zero-shot and few-shot performance, we observed significant improvements with few-shot examples in models like Tulu2, Mixtral, and GPT-3.5 (0613). However, Llama-3 exhibits similar performance in both zero-shot and few-shot settings. Interestingly, few-shot samples improves Llama-3's performance for discharge summaries and physician notes, but degrades for nursing notes. This suggests that Llama-3 struggles to derive general patterns from in-context examples, particularly in more unstructured formats. Discharge summaries and physician notes typically contain semi-structured patterns (*e.g.*, "*[2022-02-02 04:06:00] WBC - 9.6*"), making it easier for models to generalize from in-context examples. In contrast, nursing notes are often written in free-form text, presenting a challenge for Llama-3 to generalize effectively from few-shot samples.

---

[10]We provided GPT-3.5 (0613) with all the necessary information (*e.g.,* column name, table name, column descriptions) for verification and conducted the consistency check directly.

### 5.3 Result in MIMIC-OMOP

The MIMIC-III database stores various clinical events in multiple tables like *"chartevents"* and *"labevents"*, while the OMOP CDM organizes these events into standardized tables such as the *"measurements"* table, facilitating multi-organization biomedical research. Additionally, MIMIC-III uses a variety of dictionary tables (*e.g.*, d_items, d_icd_diagnoses), while the OMOP CDM uses only the *"concept"* table for this purpose (see Appendix B.2.1). This characteristic of the OMOP CDM simplifies table identification and item search within the database, leading us to anticipate superior performance of MIMIC-OMOP over MIMIC-III. Contrary to our expectations, however, the performance on MIMIC-OMOP was found to be similar to or lower than that of MIMIC, as shown in Table 3. To understand this discrepancy, we conducted an analysis based on entity types (see Sec. 3.1) and discovered that a significant performance drop occurred with Type 1 entities. This decline was mainly caused by the complexity of entities within the MIMIC-OMOP database. MIMIC-OMOP includes detailed and diverse information related to specific entity names, such as *"Cipralex 10mg tablets (Sigma Pharmaceuticals Plc) 28 tablets"*, encompassing value, unit, and other related details all at once. Our findings indicate the necessity for developing a framework that can freely interact with the database to overcome these challenges in future research. The detailed experimental results for each type of OMOP and MIMIC-III can be found in Appendix M.

### 5.4 Component Analysis

For evaluating the role of each component of our framework, we performed further analysis on 25% of the entire test set. This analysis involved three distinct experimental settings. In the first experimental setting, we excluded the NER stage (Figure 3-2) and provided the ground truth entities. The second setting built upon this by adding ground truth for the time filtering and table identification (Figure 3-3,4) stages. In the third setting, we further included ground truth for pseudo table creation, self correction, and value reformatting stage (Figure 3-5,6,7). According to Table 10, our experiments with GPT-3.5 (0613) demonstrated a significant improvement in recall: 76.11% at the first setting, 82.49% at the second, and 92.83% at the third, with an approximate increase of 8 percentage points at each setting. This finding indicates that the information provided at each stage plays a crucial role in enabling the model to better understand and solve the task. Notably, the performance in the third setting exceeded 92%, showing a significant improvement over the second setting, indicating that LLMs struggle considerably with converting free text into a structured format. Refer to Appendix N for experimental results and additional analysis.

## 6 Conclusion and Future Direction

In this paper, we introduce `EHRCon`, a carefully crafted dataset designed to improve the accuracy and reliability of EHRs. By meticulously comparing clinical notes with their corresponding database, `EHRCon` addresses critical inconsistencies that can jeopardize patient safety and care quality. Alongside `EHRCon`, we present CheckEHR, an innovative framework that leverages LLMs to efficiently verify data consistency within EHRs. Our study lays the groundwork for future advancements in automated and dependable healthcare documentation systems, ultimately enhancing patient safety and streamlining healthcare processes.

Despite the careful design of our dataset, several limitations exist. First, although MIMIC-III is hospital data, preprocessing is required to protect patient privacy. This preprocessing can introduce inconsistencies that do not occur in the actual hospital setting. Therefore, the inconsistencies we identified may not be present in real hospital data. In this regard, future research should incorporate consistency checks using real hospital data to identify inconsistency patterns in practical settings. Secondly, despite the high quality of our dataset, created by highly trained human annotators, there are limitations in verifying the contents of all clinical notes in MIMIC-III. To cover a broader range of cases, more scalable methods will be required.

## Acknowledgments and Disclosure of Funding

This work was supported by the Institute of Information & Communications Technology Planning & Evaluation (IITP) grant (No.RS-2019-II190075), National Research Foundation of Korea (NRF) grant

(NRF-2020H1D3A2A03100945), and the Korea Health Industry Development Institute (KHIDI) grant (No.HR21C0198, No.HI22C0452), funded by the Korea government (MSIT, MOHW).

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
