# Supplementary Contents

# A  Datasheet for Datasets

## A.1  Motivation

- **For what purpose was the dataset created?**

  EHRs are integral for storing comprehensive patient medical records, combining structured data with detailed clinical notes. However, they often suffer from discrepancies due to unintuitive EHR system designs and human errors, posing serious risks to patient safety. To address this, we developed `EHRCon`.

- **Who created the dataset (e.g., which team, research group) and on behalf of which entity (e.g., company, institution, organization)?**

  The authors created the dataset.

- **Who funded the creation of the dataset? If there is an associated grant, please provide the name of the grantor and the grant name and number.**

  This work was supported by the Institute of Information & Communications Technology Planning & Evaluation (IITP) grant (No.RS-2019-II190075), National Research Foundation of Korea (NRF) grant (NRF-2020H1D3A2A03100945), and the Korea Health Industry Development Institute (KHIDI) grant (No.HR21C0198, No.HI22C0452), funded by the Korea government (MSIT, MOHW).

## A.2  Composition

- **What do the instances that comprise the dataset represent (e.g., documents, photos, people, countries)?**

  `EHRCon` includes entities identified in the notes along with their labels. Additionally, for inconsistent entities, it identifies the specific table and column in the EHR where the inconsistencies are found.

- **How many instances are there in total (of each type, if appropriate)?**

  It includes 4,101 entities extracted from a total of 105 clinical notes.

- **Does the dataset contain all possible instances or is it a sample (not necessarily random) of instances from a larger set?**

  We randomly extracted and used 105 clinical notes from MIMIC-III. The notes consist of discharge summaries, physician notes, and nursing notes.

- **What data does each instance consist of?**

  Entities extracted from the notes have corresponding labels, and for inconsistent entities, the specific tables and columns in the EHR where the inconsistency occurs are recorded.

- **Is there a label or target associated with each instance?**

  Each entity has a corresponding label.

- **Is any information missing from individual instances? If so, please provide a description, explaining why this information is missing (e.g., because it was unavailable). This does not include intentionally removed information, but might include, e.g., redacted text.**

  No.

- **Are relationships between individual instances made explicit (e.g., users' movie ratings, social network links)?**

  No.

- **Are there recommended data splits (e.g., training, development/validation, testing)?**

  We randomly divided 105 clinical notes into a test set of 83 and a validation set of 22 for the experiment.

- **Are there any errors, sources of noise, or redundancies in the dataset?**

  Although trained human annotators followed the labeling instructions, slight variations may exist due to individual perspectives.

- **Is the dataset self-contained, or does it link to or otherwise rely on external resources (e.g., websites, tweets, other datasets)?**

EHRCon depends on MIMIC-III which is accessible via PhysioNet[11].

- **Does the dataset contain data that might be considered confidential (e.g., data that is protected by legal privilege or by doctor-patient confidentiality, data that includes the content of individuals' non-public communications)?**
  No.

- **Does the dataset contain data that, if viewed directly, might be offensive, insulting, threatening, or might otherwise cause anxiety?**
  No.

- **Does the dataset relate to people?**
  Yes.

- **Does the dataset identify any subpopulations (e.g., by age, gender)?**
  No.

- **Does the dataset contain data that might be considered sensitive in any way (e.g., data that reveals race or ethnic origins, sexual orientations, religious beliefs, political opinions or union memberships, or locations; financial or health data; biometric or genetic data; forms of government identification, such as social security numbers; criminal history)?**
  No.

## A.3 Collection process

- **How was the data associated with each instance acquired?**
  Before starting the labeling process, we designed the labeling instructions in consultation with the partitioners. Based on these instructions, trained human annotators reviewed the clinical notes and labeled the entities.

- **What mechanisms or procedures were used to collect the data (e.g., hardware apparatuses or sensors, manual human curation, software programs, software APIs)?**
  We used Google Search and ChatGPT4 API (Azure) to analyze the entities in clinical notes, and SQLite3 for SQL query execution.

- **If the dataset is a sample from a larger set, what was the sampling strategy (e.g., deterministic, probabilistic with specific sampling probabilities)?**
  When extracting notes, we randomly selected notes with at least 800 tokens to ensure they contained sufficient content.

- **Who was involved in the data collection process (e.g., students, crowd workers, contractors) and how were they compensated (e.g., how much were crowd workers paid)?**
  The authors manually labeled the data.

- **Over what timeframe was the data collected?**
  We created the dataset from April 2023 to April 2024.

- **Were any ethical review processes conducted (e.g., by an institutional review board)?**
  N/A.

- **Does the dataset relate to people?**
  Yes.

- **Did you collect the data from the individuals in question directly, or obtain it via third parties or other sources (e.g., websites)?**
  N/A.

- **Were the individuals in question notified about the data collection?**
  N/A.

- **Did the individuals in question consent to the collection and use of their data?**
  N/A.

---

[11]https://physionet.org/

- **If consent was obtained, were the consenting individuals provided with a mechanism to revoke their consent in the future or for certain uses?**

  N/A.

- **Has an analysis of the potential impact of the dataset and its use on data subjects (e.g., a data protection impact analysis) been conducted?**

  Yes.

## A.4 Preprocessing/cleaning/labeling

- **Was any preprocessing/cleaning/labeling of the data done (e.g., discretization or bucketing, tokenization, part-of-speech tagging, SIFT feature extraction, removal of instances, processing of missing values)?**

  We performed preprocessing by removing pre-admission records and treatment plans from the clinical notes.

- **Was the "raw" data saved in addition to the preprocess/cleaned/labeled data (e.g., to support unanticipated future uses)?**

  We additionally provide labels for the original notes.

- **Is the software that was used to preprocess/clean/label the data available?**

  We performed preprocessing using Python.

## A.5 Uses

- **Has the dataset been used for any tasks already?**

  No.

- **Is there a repository that links to any or all papers or systems that use the dataset?**

  N/A.

- **What (other) tasks could the dataset be used for?**

  It can be used not only for consistency checks of EHR but also for table-based fact verification tasks.

- **Is there anything about the composition of the dataset or the way it was collected and preprocessed/cleaned/labeled that might impact future uses?**

  N/A.

- **Are there tasks for which the dataset should not be used?**

  N/A.

## A.6 Distribution

- **Will the dataset be distributed to third parties outside of the entity (e.g., company, institution, organization) on behalf of which the dataset was created?**

  No.

- **How will the dataset be distributed?**

  The dataset will be released at PhysioNet.

- **Will the dataset be distributed under a copyright or other intellectual property (IP) license, and/or under applicable terms of use (ToU)?**

  The dataset is released under MIT License.

- **Have any third parties imposed IP-based or other restrictions on the data associated with the instances?**

  No.

- **Do any export controls or other regulatory restrictions apply to the dataset or to individual instances?**

  No.

**A.7 Maintenance**

- **Who will be supporting/hosting/maintaining the dataset?**
  The authors will support it.

- **How can the owner/curator/manager of the dataset be contacted(e.g., email address)?**
  Contact the authors ({yeonsu.k, jiho.kim}@kaist.ac.kr).

- **Is there an erratum?**
  No.

- **Will the dataset be updated (e.g., to correct labeling erros, add new instances, delete instances)?**
  The authors will update the dataset if any corrections are required.

- **If the dataset relates to people, are there applicable limits on the retention of the data associated with the instances (e.g., were the individuals in question told that their data would be retained for a fixed period of time and then deleted)?**
  N/A

- **Will older versions of the dataset continue to be supported/hosted/maintained?**
  We plan to upload the latest version of the dataset and will document the updates for each version separately.

- **If others want to extend/augment/build on/contribute to the dataset, is there a mechanism for them to do so?**
  Contact the authors ({yeonsu.k, jiho.kim}@kaist.ac.kr).

## B MIMIC-III and OMOP CDM

EHRCon is built on two types of relational databases: MIMIC-III and its version in OMOP CDM. This structure allows us to incorporate various schema types and enhance generalizability. In this section, we will describe in detail the process of creating both MIMIC-III and MIMIC-OMOP.

### B.1 Data Preparation

We created EHRCon using 105 randomly selected clinical notes and 13 tables. In this section, we will provide a detailed description of how the clinical notes and tables were selected and preprocessed.

#### B.1.1 Note Preparation

To develop a more realistic dataset that mirrors a typical hospital setting, we began by analyzing the clinical notes from MIMIC-III by category (see Figure 4). The largest portion of notes came from the "Nursing/other" category. However, much of this content, such as details of family meetings, couldn't be verified against the table data and was therefore excluded. Radiology reports and ECG (electrocardiogram) reports were also excluded since they rely on imaging and cardiac monitoring, which are outside our scope. Therefore, our focus was on discharge summaries, nursing notes, and physician notes, as these are commonly used in hospitals and related to tabular information.

Additionally, clinical notes may contain pre-admission history and future treatment plans not found in the EHR tables. To concentrate on current admission records, we filtered out this additional information from the notes. To ensure sufficient detail, we randomly selected 105 notes with more than 800 tokens for the experiments.

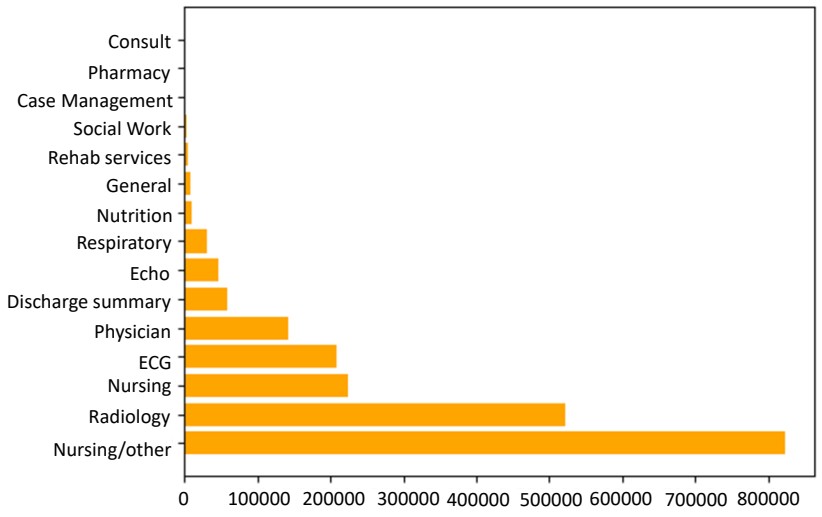

Figure 4: Distribution of different note categories in MIMIC-III.

#### B.1.2 Table Preparation

To enhance the utility of our dataset, we identified tables in the MIMIC-III database that contain entities from discharge summaries, physician notes, and nursing notes. To achieve this, we analyzed randomly selected 300 clinical notes, consisting of 100 discharge summaries, 100 nursing notes, and 100 physician notes. As a result, we concluded with a total of thirteen tables, including four dictionary tables, as follows: Chartevents, Labevents, Prescriptions, Inputevents_cv, Inputevents_mv, Outputevents, Microbiologyevents, Diagnoses_icd, Procedures_icd, D_items, D_icd_diagnoses, D_icd_procedures, and D_labitems. The overall table preparation process is described in Figure 5.

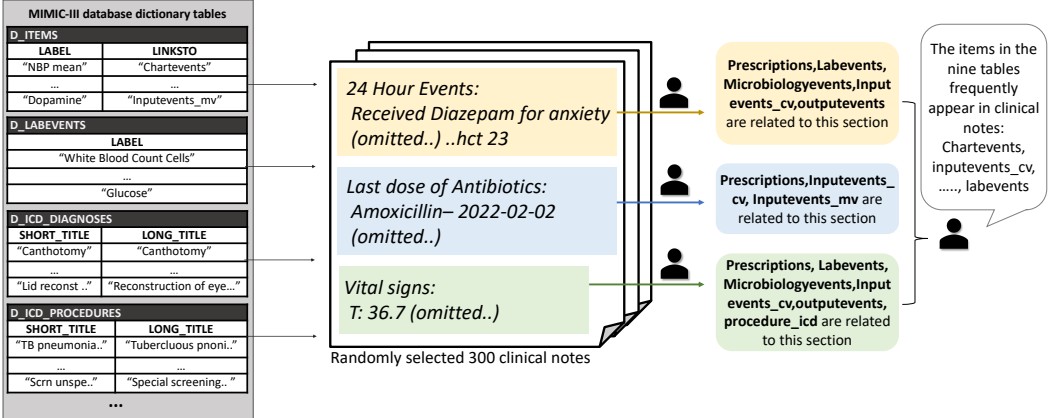

Figure 5: Overall process of table preparation.

## B.2 OMOP CDM

### B.2.1 Matching OMOP CDM and MIMIC-III

Table 4 illustrates how each MIMIC-III table and column correspond to the respective OMOP CDM table and column.

### B.2.2 Labeling Process of MIMIC-OMOP

MIMIC-OMOP is derived from MIMIC-III, and while both are organized using different table structures (schemas), they contain the same patient information. Therefore, the labels of the entities annotated in MIMIC-III can be directly applied to MIMIC-OMOP as well.

We downloaded the MIMIC-OMOP database[12], which follows the mapping guidelines specified in Section B.2.1. Upon reviewing MIMIC-OMOP, we discovered that the drug exposure start time was recorded as being later than the end time. We corrected this issue by swapping the values in the columns.

## C Labeling Instructions

### C.1 Task Description

Identify the entities mentioned in the clinical note and write an SQLite3 query to check if the corresponding values exist in the database. If you encounter any ambiguous or corner cases during the annotation process, make a note of them and discuss them with the practitioners and other annotators.

### C.2 Identify the Entity

Identify the entities that likely exist in the following tables: Procedures_icd, Diagnoses_icd, Microbiologyevents, Chartevents, Labevents, Prescriptions, Inputevents_mv, Inputevents_cv, and Outputevents. Detect entities that occurred exclusively between the patient's admission and the charted time of the note.

### C.2.1 Additional Rules (for Discharge Summary)

- For diagnoses, extract entities from the Discharge Diagnosis list, including both the *Primary Diagnosis* and *Secondary Diagnosis* sections.
- For procedures, extract entities from both the *Major Surgical* and *Invasive Procedure* sections.

---

[12]Database sourced from `https://github.com/MIT-LCP/mimic-omop`.

Table 4: Relationship between MIMIC-III and MIMIC-OMOP Tables.

| MIMIC-III | | MIMIC-OMOP | |
|---|---|---|---|
| **Table** | **Column** | **Table** | **Column** |
| Chartevents | chartttime | Measurement | measurement_datetime |
| | valuenum | | value_as_number |
| | valueuom | | unit_source_value |
| Labevents | chartttime | | measurement_datetime |
| | valuenum | | value_as_number |
| | valueuom | | unit_source_value |
| Outputevents | chartttime | | measurement_datetime |
| | valuenum | | value_as_number |
| | valueuom | | unit_source_value |
| Microbiologyevents | charttime | | measurement_datetime |
| | | Specimen | specimen_datetime |
| | org_name | Concept | concept_name |
| | spec_type_desc | | |
| Inputevents_mv | starttime | Drug_exposure | drug_exposure_startdate |
| | endtime | | drug_exposure_enddate |
| | amount | | - |
| | amoutuom | | - |
| | rate | | - |
| | rateuom | | - |
| Inputevents_cv | charttime | | drug_exposure_startdate drug_exposure_enddate |
| | amount | | - |
| | amoutuom | | - |
| | rate | | - |
| | rateuom | | - |
| Prescriptions | startdate | | drug_exposure_startdate |
| | enddate | | drug_exposure_enddate |
| | dose_val_rx | | - |
| | dose_unit_rx | | - |
| | drug | Concept | concept_name |
| D_items | label | | |
| D_labitems | label | | |
| D_icd_diagnosis | short_title | | |
| | long_title | | |
| D_icd_procedures | short_title | | |
| | long_title | | |

## C.3 Classify the Type of the Identified Entity

- Type 1: Entities with numerical values.
  - When the value associated with the entity is numeric.
    * *e.g.*, *temp - 99.9, WBC - 9.6*
- Type 2: Entities without numerical values but whose existence can be verified in the database.
  - When the value associated with the entity is not numeric and it is sufficient to confirm its existence without checking the value in the database.
    * *e.g.*, *Lasix was started, WBC was tested*
- Type 3: Entities with string values.
  - When the value associated with the entity is not numeric and it is necessary to check the value in the database.
    * *e.g.*, *Lasix increased, BP was stable, WBC changed from 1 to 5, temp > 100*

## C.4 Search Items in the Database

Use the Item Search Tool to find items in the database related to the detected entity. From these, select those that accurately represent the detected entity.

- Guidelines for using the Item Search Tool
  - Display items related to the entity in the following order: D_labitems, D_items, Prescriptions, D_icd_procedures, D_icd_diagnoses
  - If none of the searched items can accurately represent the entity, manually search for the items in the database and add it.

## C.5 Select Tables and Enter the Evidence Line

- Check the tables connected to the selected items and select the tables to which the detected entity can be linked.
- Enter the line number in the clinical notes where the entity is located.

## C.6 Check the Number of Entities

This section is for checking how many times an entity is mentioned. For example, in the case of BP, if it is written as *120/50*, it should be considered as two mentions of BP. Each instance should then be verified to ensure it is correctly recorded in the database.

## C.7 Extract Information Related to the Entity from the Clinical Note

Extract the value, unit, time, organism, and specimen related to the entity. Refer to Figure 6 for the actual table columns corresponding to value, unit, time, organism and specimen.

- Value: Numeric value corresponding to the entity (*e.g.*, 184, 103.3).
- Unit: Unit corresponding to the numeric value (*e.g.*, mg, ml/h). If the unit is not specified, do not extract it.
- Time of the clinical event occurrence
  - If only the date is noted, use the format YYYY-MM-DD. If only MM-DD is noted, use the year from the note's chartdate to complete as YYYY-MM-DD.
  - In case the time is also specified, use the format YYYY-MM-DD HH:MM:SS (24-hour format).
  - For relative time expression (*e.g.*, *HD #3*), calculating the date based on the admission date. For '*Yesterday*', calculate it based on the note's chartdate.
  - If 'admission', 'charttime', or 'discharge' is noted, calculate the date based on the respective entry.

- If there is no time noted, for discharge summaries, consider the entire hospitalization period. For nursing and physician notes, consider the period within one day before and one day after the chartdate.

- Organism: This corresponds to a microbiology event and involves microorganism (*e.g.*, *bacteria, fungi*).

- Specimen: This corresponds to a microbiology event and involves specimen (*e.g.*, *blood, urine, tissue*) from which the microbiological sample was obtained.

---

{'**value**': {'labevents': 'valuenum', 'chartevents': 'valuenum', 'inputevents_cv': 'amount', `inputevents_cv`: `rate`, 'outputevents': 'valuenum', 'prescriptions': 'dose_val_rx', 'inputevents_mv': 'rate', 'inputevents_mv': 'amount'}

{'**unit**': {'chartevents': 'valueuom', 'inputevents_cv': 'amountuom', 'inputevents_cv': 'rateuom', 'inputevents_mv': 'rateuom', 'inputevents_mv ': 'amoutuom', 'labevents': 'valueuom', 'outputevents': 'valueuom', 'prescriptions': 'dose_unit_rx'}

{'**time**': {'chartevents': 'charttime', 'labevents': 'charttime', 'outputevents': 'charttime', 'inputevents_cv': 'charttime', 'microbiologyevents': 'charttime', 'inputevents_mv': 'starttime', 'inputevents_mv': 'endtime', 'prescriptions': 'startdate', 'prescriptions': 'enddate'}

{'**organism**': {'microbiologyevents': 'org_name'},

{'**specimen**': {'microbiologyevents': 'spec_type_desc'}}

---

Figure 6: Actual table columns corresponding to value, unit, time, organism, and specimen.

## C.8  Example of Annotation

- Physical Exam: Temperature 99.9
  - Label based on whether there is a record with a value of 99.9 in the 'Temperature' data.
- Oxygen saturation 98%
  - Label based on whether there is a record with a value of 98 and unit of % in the 'Oxygen saturation' data.
- BP: 120/80 (90)
  - Label whether 'BP' has the records of 120, 80, and 90.

## C.9  Query Generation

Create an SQL query that utilizes the values extracted from the clinical note and satisfies the conditions specified in Figure 8. If an inconsistency occurs, mask the condition values, execute the query on the database, and identify the table and columns causing the inconsistency.

## C.10  Notice

- For blood pressure, if given in the format (num1/num2), each of num1 and num2 should be checked individually.
- Exclude entities that are not explicitly identified (*e.g.*, Chem 7: 140 / 4.2 / 104 / 25 / 15 / 1.0 / 90).

# D  Time Expressions in Clinical Notes

Figure 7 provides examples categorized by type of time expression. Furthermore, Figure 8 depicts the verification ranges for various table types based on these time expressions.

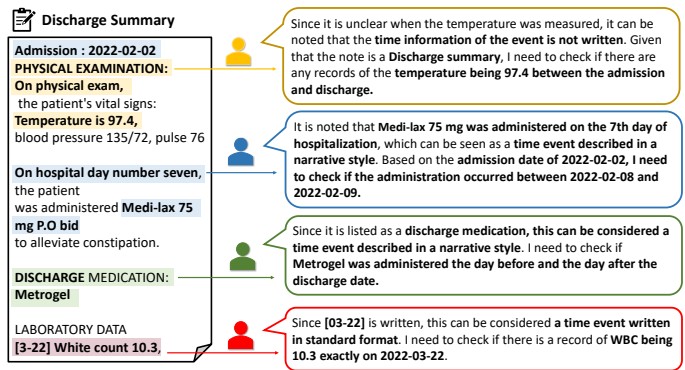

Figure 7: Examples categorized by type of time expression.

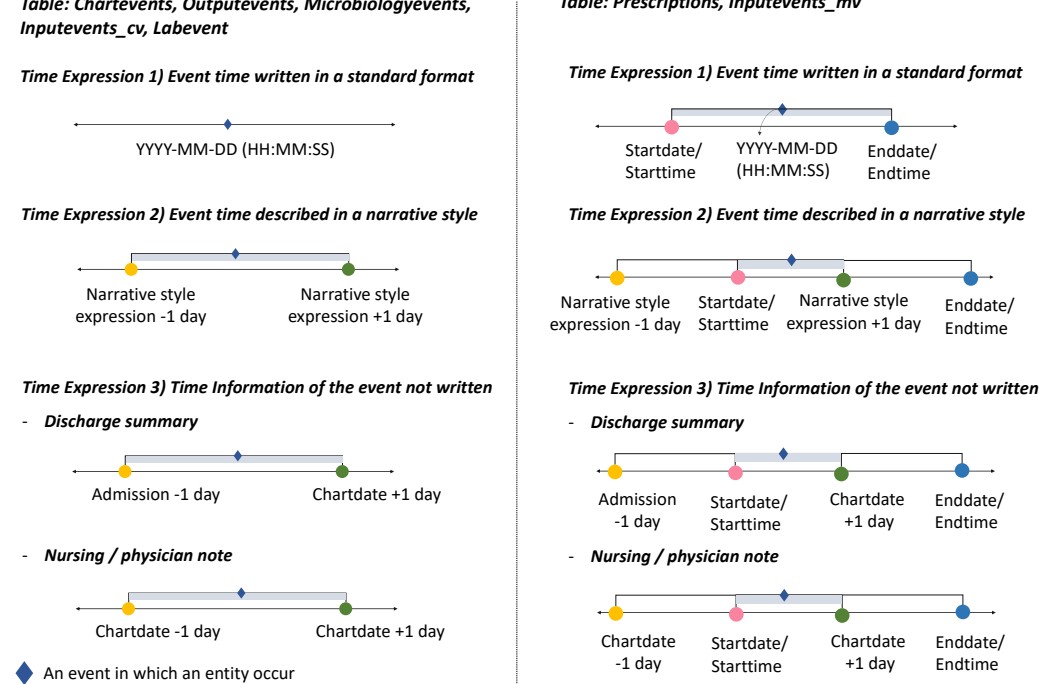

Figure 8: Range of verification based on temporal expressions. The left side indicates time solely through 'charttime'. On the right, however, 'prescriptions' and 'inputevents_mv' utilize both 'start-time' and 'endtime', creating a varied verification range for time.

# E Quality Control

To ensure the high quality of our dataset, we rely on expert researchers for annotation rather than crowd-sourced workers. The researchers have published AI healthcare-related papers and possess a thorough understanding of the MIMIC-III database and SQL syntax. After the labeling was completed, four annotators conducted cross-validation on a total of 20 notes.[13] As a result, the F1 score of the entities recognized by the annotators was 0.880. Additionally, among the entities recognized by both annotators, the cases where the labels were the same accounted for 0.938. This demonstrates the consistency of our labeling process.

---

[13]Each type of note (*e.g.*, *discharge summary, physician note, nursing note*) has its own unique format and style. Therefore, we manually cross-checked only 20 notes for each type, focusing on these characteristics. For the remaining notes, we corrected the data based on the key elements resolved during the cross-check to maintain consistency.

## F   Error cases

We conducted an in-depth analysis of various discrepancies observed in EHRCon. To achieve this, we compared the coverage of tables where discrepancies occur in EHRCon and analyzed the error rates per column. Additionally, we examined in detail which columns have higher error occurrence rates for each note. These details can be found in Figure 9 to 12.

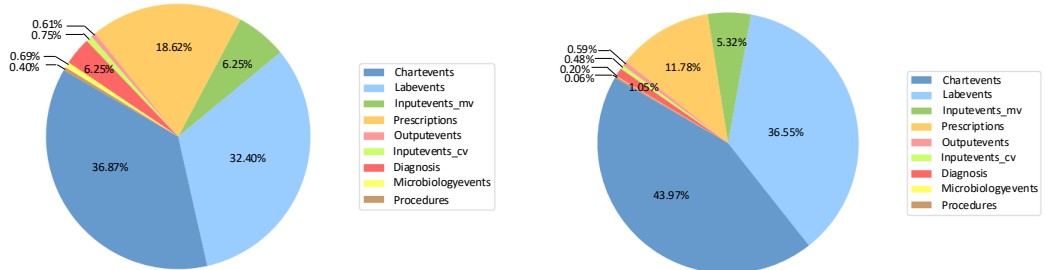

Figure 9: Proportion of tables used in EHRCon, covering various clinical events in a hospital such as vital signs, medications, and lab results.

Figure 10: Proportion of tables with inconsistencies. Every table exhibited inconsistencies, with the highest rate observed in labevents.

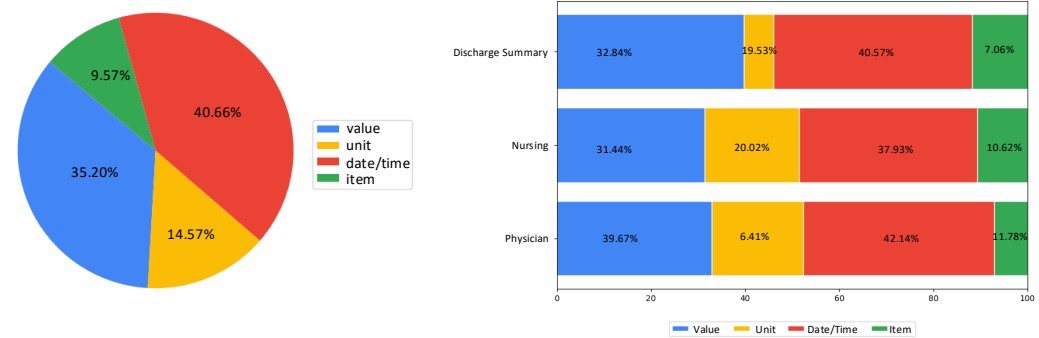

Figure 11: Proportion of columns with inconsistencies. The highest frequency of inconsistencies occurred in the date and time columns.

Figure 12: Proportion of inconsistencies by column in each note. Unlike discharge summaries and physician notes, nursing notes had the highest occurrence of inconsistencies within the unit.

## G   Prompt

The prompts used in CheckEHR can be found in Figure 13 to 20. To protect patient privacy, all values in the example data mentioned in the paper have been replaced with fictional values. Additionally, some sentences have been paraphrased or omitted.

In particular, table descriptions and column descriptions were utilized to effectively conduct the table identification and pseudo table creation processes. The descriptions used for this purpose, derived using MIMIC-III documentation[14] and ChatGPT-4, can be found in Table 5 and Table 6.

---

[14]https://mimic.mit.edu/docs/iii/

Table 5: Table description used in the Table Identification prompt.

| Database | Table | Description |
|---|---|---|
| MIMIC-III | D_items | D_items provides metadata for all recorded items, including medications, procedures, and other clinical measurements, with unique identifiers, labels, and descriptions. |
| MIMIC-III | Chartevents | Chartevents contains time-stamped clinical data recorded by caregivers, such as vital signs, laboratory results, and other patient observations, with references to the D_items table for item details. |
| MIMIC-III | Inputevents_cv | Inputevents_cv contains detailed data on all intravenous and fluid inputs for patients during their stay in the ICU and uses ITEMID to link to D_items. |
| MIMIC-III | Inputevents_mv | The inputevents_mv table records detailed information about medications and other fluids administered to patients, including dosages, timings, and routes of administration, specifically from the MetaVision ICU system. |
| MIMIC-III | Microbiologyevents | Microbiologyevents contains detailed information on microbiology tests, including specimen types, test results, and susceptibility data for pathogens identified in patient samples. This information is linked to D_items by ITEMID. |
| MIMIC-III | Outputevents | Records information about fluid outputs from patients, such as urine, blood, and other bodily fluids, including timestamps, amounts, and types of outputs, with references to the D_items table for item details. |
| MIMIC-III | D_labitems | D_labitems contains metadata about laboratory tests, including unique identifiers, labels, and descriptions for each lab test performed. |
| MIMIC-III | Labevents | Labevents contains detailed records of laboratory test results, including test values, collection times, and patient identifiers, with references to the D_labitems table for test-specific metadata. |
| MIMIC-III | Prescriptions | Lists patient prescriptions with details on dose, administration route, and frequency. There is no reference table. |
| MIMIC-III | D_icd_diagnoses | The D_icd_diagnoses table provides descriptions and categorizations for ICD diagnosis codes used to classify patient diagnoses. |
| MIMIC-III | Diagnoses_icd | The Diagnoses_icd table contains records of ICD diagnosis codes assigned to patients, linking each diagnosis to specific hospital admissions. |
| MIMIC-III | D_icd_procedures | D_icd_procedures contains definitions and details for ICD procedure codes, including code descriptions and their corresponding categories. |
| MIMIC-III | Procedures_icd | Procedures_icd records the procedures performed on patients during their hospital stay, indexed by ICD procedure codes and linked to specific hospital admissions. |
| MIMIC-OMOP | Concept | The concept table is a standardized lookup table that contains unique identifiers and descriptions for all clinical and administrative concepts, providing a consistent way to reference data elements across various healthcare domains. |
| MIMIC-OMOP | Measurement | Table stores quantitative or qualitative data obtained from tests, screenings, or assessments of a patient, including lab results, vital signs, and other measurable parameters related to patient health. |
| MIMIC-OMOP | Durg_exposure | Drug_exposure table records details about the dispensing and administration of drugs to a patient, including the drug type, dosage, route, and duration of each drug exposure event. |
| MIMIC-OMOP | Specimen | Specimen table contains details about patient specimens, including type, collection method, and collection context. |
| MIMIC-OMOP | Condition_occurrence | The table logs instances of clinical conditions diagnosed or reported in a patient, detailing the type of condition, diagnosis date, and source information. |
| MIMIC-OMOP | Procedure_occurrence | Procedure_occurrence table captures details of medical procedures performed on a patient, including the type of procedure, date, and relevant context from the healthcare encounter. |

Table 6: Column description used in the Pseudo Table Creation prompt.

| Database | Table | Column | Description |
|---|---|---|---|
| MIMIC-III | D_items | Label | The label column provides a human-readable name for this item. The label could be a description of a clinical observation such as Temperature, blood pressure, and heart rate. |
| MIMIC-III | D_labitems | Label | The label column provides a human-readable name for this item. The label could be a description of a clinical observation such as wbc, glucose, and PTT. |
| MIMIC-III | D_icd_procedures | Short_title | Short_title provides a concise description of medical procedures encoded by ICD-9-CM codes. |
| MIMIC-III | D_icd_procedures | Long_title | Long_title offers a detailed and comprehensive description of medical procedures associated with ICD-9-CM codes. |
| MIMIC-III | D_icd_diagnoses | Short_title | Short_title provides brief descriptions or names of medical diagnoses corresponding to their ICD-9 codes. |
| MIMIC-III | D_icd_diagnoses | Long_title | Long_title offers more detailed descriptions of medical diagnoses corresponding to their ICD-9 codes. |
| MIMIC-III | Chartevents | Charttime | Charttime records the time at which an observation occurred and is usually the closest proxy to the time the data was measured, such as admission time or a specific date like 2112-12-12. |
| MIMIC-III | Chartevents | Valuenum | This column contains the numerical value of the laboratory test result, offering a quantifiable measure of the test outcome. If this data is not numeric, Valuenum must be null. |
| MIMIC-III | Chartevents | Valueuom | Valueuom is the unit of measurement. |
| MIMIC-III | Inputevents_cv | Charttime | Charttime represents the time at which the measurement was charted. |
| MIMIC-III | Inputevents_cv | Amount | Indicates the total quantity of the input given during the charted event. |
| MIMIC-III | Inputevents_cv | Amountuom | Amountuom is the unit of Amount. |
| MIMIC-III | Inputevents_cv | Rate | Details the rate at which the input was administered, typically relevant for intravenous fluids or medications. |

Table 6: Column description used in the Pseudo Table Creation prompt. (Continued)

| Database | Table | Column | Description |
|---|---|---|---|
| MIMIC-III | Inputevents_cv | Rateuom | Rateuom is the unit of Rate. |
| MIMIC-III | Labevents | Charttime | The Charttime column records the exact timestamp when a laboratory test result was charted or documented (*e.g.*, *2112-12-12*). |
| MIMIC-III | Labevents | Valuenum | The Valuenum column contains the numeric result of a laboratory test, represented as a floating-point number. |
| MIMIC-III | Labevents | Valueuom | The valueuom column specifies the unit of measurement for the numeric result recorded in the Valuenum column. |
| MIMIC-III | Inputevents_mv | Starttime | The Starttime column records the timestamp indicating when the administration of a medication or other clinical intervention was initiated. |
| MIMIC-III | Inputevents_mv | Endtime | The Endtime column records the timestamp indicating when the administration of a medication or other clinical intervention was completed. |
| MIMIC-III | Inputevents_mv | Amount | Amount records the total quantity of a medication or fluid administered to the patient. |
| MIMIC-III | Inputevents_mv | Amountuom | Amountuom specifies the unit of measurement for the amount of medication or fluid administered, such as milliliters (ml) or milligrams (mg). |
| MIMIC-III | Inputevents_mv | Rate | Rate specifies the rate at which a medication or fluid was administered. |
| MIMIC-III | Inputevents_mv | Rateuom | Rateuom specifies the unit of measurement for the rate at which a medication or fluid was administered, such as milliliters per hour (mL/hr). |
| MIMIC-III | Outputevents | Charttime | Charttime records the timestamp when an output event, such as urine output or drainage, was documented. |
| MIMIC-III | Outputevents | Valuenum | Valuenum contains the numeric value representing the quantity of output, such as the volume of urine or other fluids, recorded as a floating-point number. |
| MIMIC-III | Outputevents | Valueuom | Valueuom specifies the unit of measurement for the numeric value recorded in the Valuenum column, such as milliliters (mL). |
| MIMIC-III | Prescriptions | Startdate | The Startdate column records the date when a prescribed medication was first ordered or administered to the patient. |
| MIMIC-III | Prescriptions | Enddate | The Enddate column records the date when the administration of a prescribed medication was completed or discontinued. |
| MIMIC-III | Prescriptions | Drug | The Drug column lists the name of the medication that was prescribed to the patient. |
| MIMIC-III | Prescriptions | Dose_val_rx | The Dose_val_rx column specifies the numeric value of the prescribed dose for the medication. |
| MIMIC-III | Prescriptions | Dose_unit_rx | The Dose_unit_rx column specifies the unit of measurement for the prescribed dose of the medication, such as milligrams (mg) or milliliters (mL). |
| MIMIC-III | Microbiologyevents | Charttime | The Charttime column records the timestamp when the microbiological culture result was documented or charted. |
| MIMIC-III | Microbiologyevents | Org_name | The Org_name column identifies the name of the organism *(such as a bacterium or fungus)* that was detected in a microbiological culture. |
| MIMIC-III | Microbiologyevents | Spec_type_desc | The Spec_type_desc column describes the type of specimen *(such as blood, urine, or tissue)* from which the microbiological culture was obtained. |
| MIMIC-OMOP | Concept | Concept_name | The Concept_name column contains an unambiguous, meaningful, and descriptive name for the Concept. |
| MIMIC-OMOP | Drug_exposure | Drug_exposure _start_date | The start date for the current instance of Drug utilization. Valid entries include a start date of a prescription, the date a prescription was filled, or the date on which a Drug administration procedure was recorded. |
| MIMIC-OMOP | Drug_exposure | Drug_exposure _end_date | The end date for the current instance of Drug utilization. It is not available from all sources. |
| MIMIC-OMOP | Drug_exposure | Quantity | The quantity column records the amount of the drug administered or prescribed, providing essential information for dosage and treatment analysis. |
| MIMIC-OMOP | Drug_exposure | Dose_unit _source_value | The dose_unit_source_value captures the original unit of measurement for the drug dosage as recorded in the source data, preserving the raw data detail for reference and mapping purposes. |
| MIMIC-OMOP | Measurements | Measurement _datetime | The measurement_datetime records the exact date and time when the measurement was taken, providing precise temporal context for each measurement entry. |
| MIMIC-OMOP | Measurements | Value_as _number | The value_as_number stores the numerical result of the measurement, allowing for quantitative analysis of the recorded data. |
| MIMIC-OMOP | Measurements | Unit_source _value | The unit_source_value captures the original unit of measurement as recorded in the source data, preserving the context of the measurement's unit before any standardization. |
| MIMIC-OMOP | Specimen | Specimen _datetime | The specimen_datetime records the exact date and time when the specimen was collected, providing precise temporal context for each specimen entry. |

## Prompt Template for Note Segmentation

Task : Your task is to analyze a clinical note and divide it into three sections based on thematic or semantic coherence. Each section should center around a unique theme or idea, providing a cohesive view of the content.

Please follow these guidelines:

- Thematic or Semantic Unity: Group content based on clear thematic or semantic relationships, ensuring that each section covers a distinct aspect related to the overall topic. Everything within a section should be related and contribute to a unified understanding of that theme.

- Equal Length and Comprehensive Coverage: Strive for a balance in the length of each section, but also consider the depth and breadth of the content. The division should reflect an equitable distribution of information, without sacrificing the completeness of any thematic area. This balance is essential to ensure no single section is overwhelmingly long or short compared to the others.

- Integrity of Sections: Pay close attention to the natural divisions within the text, such as headings or topic changes (*e.g.*, '*History of Present Illness*'). Ensure that these content blocks are not fragmented across sections. A section should encompass complete thoughts or topics to preserve the logical flow and coherence of information.

- Completeness of Sentences: When dividing the note, ensure each section ends with complete sentences, preventing sentences from being split across sections.

- Output format must be [section1: (start_line_number-end_line_number), section2: (start_line_number-end_line_number), section3: (start_line_number-end_line_number)].

- Precise Output Format and Continuous Line Coverage: The start line number in section1 should match the start number of the given text and the end line number in section3 should match the last line number of the given text. Ensure sections are contiguous; the end of one section immediately precedes the start of the next, with no gaps or overlaps.

Example 1)
Clinical note:
"44. On transfer, vitals were T 97.8, HR 72, BP 118/76, RR
45. She was alert, but denied dizziness, headache, palpitations,leg pain.
46. She reported mild shortness of breath, dry cough, mild nausea, lower abdominal
47. discomfort dry cough, mild nausea, lower abdominal discomfort, and decreased urine.
...
70. MEDICATIONS ON DISCHARGE:
71. 8. Zoloft 50 mg q.d.
72. 9. Nexium 20 mg b.i.d.
73. 10. Tylenol 500 mg q.d."

Output: [section1: 44-59, section2: 60-69, section3: 70-73]

Your task:
Clinical note:
"<<<<CLINICAL_NOTE>>>>"

The output format must be [section1: (start_line_number(int)-end_line_number(int)), section2: (start_line_number(int)-end_line_number(int)), section3: (start_line_number(int)-end_line_number(int))], indicating the line numbers that mark the start and end of each section.
Output: [Write your answer here]

Figure 13: Prompt Template for Note Segmentation.

## Prompt Template for Named Entity Recognition

Task : Develop a NER system to identify and categorize specific named entities in clinical texts step by step.

Guidelines: Extract entities step by step and classify entity.

- Category 1: Entities Accompanied by Numeric Values
  - Definition: This category includes entities that are mentioned along with specific numeric values. These numbers represent measurable data such as dosages, counts, measurements, etc., providing precise quantifiable information.
  - Example: '*The glucose level is 100 mg/dL*' or '*Administer 200 mg of ibuprofen*'. In these cases, the numeric values are explicitly stated.
- Category 2: Entities Mentioned Without Any Numeric Values
  - Definition: Entities that are discussed in terms of their presence, occurrence, or the fact that they were administered or performed, without providing any numerical or quantitative data, fall into this category.
  - '*The patient has been prescribed antibiotics*' or '*An MRI scan was conducted*'. Here, no specific dosage of antibiotics or quantitative results from the MRI scan are mentioned.
- Category 3: Entities with Condition-Related Information Excluding Numeric Values
  - Definition: This category captures entities related to state, condition, or outcomes that are described through qualitative assessments or descriptions without the use of explicit numeric data. It may include references to changes in condition or stability, described not with numbers but in descriptive or qualitative terms.
  - Example: '*Pt had a severe rise in ALT and AST*' or '*Pulse was dropping*'. Although these statements imply a change or assessment of condition, they do not provide specific numeric values. Instead, the focus is on qualitative descriptions of change or status, which may inherently rely on an understanding of baseline or previous values for context.

Example 1: Develop a NER system to identify and categorize specific named entities in clinical texts, step by step.
Clinical note:
"Physical Exam: Patient's t is 99. Heart shows an irregular rhythm.
...
LABORATORY DATA: On admission, WBC 11.0, hematocrit 35.0"

Step 1) Extract entities related to medication or inputevents and classify each entity. Extract the entity written in the note without modifying it.
Answer: Nothing
Step 2) Extract entities related to vital signs and classify each entity. Extract the entity written in the note without modifying it.
Answer: t - category 1 (numeric value: 99.6), heart rhythm - category 3 (qualitative assessments or descriptions: irregular)
...

Your task: Develop a NER system to identify and categorize specific named entities in clinical texts step by step.
Clinical Note: "<<<<CLINICAL_NOTE>>>>"

Figure 14: Prompt Template for Named Entity Recognition.

## Prompt Template for Time Filtering

Task: You are provided with a Discharge summary and are required to analyze time information related to an entity mentioned within the note.
Example 1) Please answer three questions that focus on {{**chloride**}}.
Clinical note:

- Admission Date: 1999-11-11

- Date charted on the note: 1999-11-13

- Content: "LABORATORY DATA: On admission, white count 9.3, hematocrit 32, platelet count 365000, PT 14, PTT 30, INR 1.3 sodium 139, potassium 4.8, {{**chloride**}} 102, CO2 26, BUN 51"

[Question 1] Based on the discharge summary provided, did the measurement for {{**chloride**}} occur during the current hospitalization period? Respond with 'Yes' if it did, or 'No' if it pertains to past medical history or conditions.
[Answer 1] Yes, because the chloride level was measured after the patient was admitted.

[Question 2] Extract and note the specific section from the discharge summary that mentions {{**chloride**}}, including any time expression associated with it. Ensure your transcription is accurate and does not infer or add details not present in the note.
[Answer 2] Note: 'LABORATORY DATA: On admission, ... {{**chloride**}} 102 ...' Time: 'admission'

[Question 3] Determine how the time of the {{**chloride**}} measurement is recorded in the note. Select the appropriate option based on the description provided:
1. Indeterminate Time stamp: Choose this if the note mentions the event in a vague or general timeframe without specific dates or times
2. Directly written in the format yyyy-mm-dd: Choose this for notes with specific dates or times in a clear, standardized format
3. Inferable Time stamp from Narrative: Choose this if the note uses terms like 'admission', 'yesterday', etc., from which the exact time of the event can be inferred based on context provided in the note.
[Answer 3] Inferable Time stamp from Narrative

Let's solve three questions that focus on {{**<<<<ENTITY>>>>**}}.

Clinical Note:

- Admission Date: <<<<ADMISSION>>>>

- Date charted on the note: <<<<CHARTTIME>>>>

- Content: "<<<<CLINICAL_NOTE>>>>"

Figure 15: Prompt Template for Time Filtering.

## Prompt Template for Table Identification

Task: Select a table based on the provided table schemas and their interconnections within the database that can store specific entity-related information. Focus on tables that are likely to contain columns relevant to the entity you are searching.

Requirements:

- Output format: [{table1, reference_table1},{table2, reference_table2},...].
- Choose from the set of table pairs: {Chartevents, D_items}, {Outputevents, D_items}, {Microbiologyevents, D_items}, {Inputevents_cv, D_items}, {Diagnoses_icd, D_icd_diagnoses}, {Procedures_icd, D_icd_procedures}, {Prescriptions}, {Inputevents_mv, D_items}, {Labevents, D_labitems}. If 'ENTITY' is not clearly recorded in any of the tables, the output should be [ 'NONE'].

Database Schema:

- D_items: A central reference table with details for items used across multiple tables, linked by ITEMID.

    - Columns: 'itemid', 'label', 'abbreviation'
    - Example Rows:
      - '1054', 'protonix', 'None'
      - '1099', 'tegretol', 'None'

- Chartevents: Includes patient observations like vital signs, uses ITEMID to link to D_items.

    - Columns: 'subject_id', 'itemid', 'charttime', 'valuenum', 'valueuom'
    - Example Rows:
      - '3', '128', '2101-10-25 04:00:00', '15.0', 'points'
      - '13', '263738', '2167-01-10 08:30:00', '84.0', 'mmHg'

Inter-table Relationships:

- Labevents is a child of D_Labitems.
- Chartevents, Inputevents_mv, Inputevents_cv, Microbiology, Outputevents are children of D_items.

Example 1) Identify the specific tables that contain definitive records of 'R'
In order to locate the 'R' within the database, examine the provided table schemas and their interconnections. Each table's purpose and the nature of the data they contain should be taken into account. Focus on the relevant columns that could store information related to 'R'. If no explicit match is found for the entity in question, indicate this with '[none]'.
Selected-Table: [chartevents, d_items]

Your task: Identify the specific tables that contain definitive records of '<<<<ENTITY>>>>'
In order to locate the '<<<<ENTITY>>>>' within the database, examine the provided table schemas and their interconnections. Each table's purpose and the nature of the data they contain should be taken into account. Focus on the relevant columns that could store information related to '<<<<ENTITY>>>>'. If no explicit match is found for the entity in question, indicate this with '[none]'.
Selected-Table: [Write your answer here]

Figure 16: Prompt Template for Table Identification.

## Prompt Template for Pseudo Table Creation

Task: The objective is to analyze a clinical note to extract specific details about an indicated Entity, focusing solely on information that is directly stated.

Instruction:

- Carefully examine the clinical note, paying close attention to any instance of the Entity highlighted as {{**Entity**}}. Focus solely on this entity for your analysis.
- Rely exclusively on the information provided within the clinical note, guided by the instructions and column descriptions provided.
- Extract and document only the information that directly pertains to the {{**Entity**}}, disregarding all other data.
- When extracting data, only include information that is explicitly mentioned in the text. Avoid making assumptions or inferring details that are not directly stated.
- Each piece of extracted information related to the Entity must be documented in the specified output format in the EHR table, detailed below, with each piece of information in separate rows: 'Mentioned [#]. DRUG: drug, STARTDATE: startdate, ENDDATE: enddate, DOSE_VAL_RX: dose_val_rx, DOSE_UNIT_RX: dose_unit_rx'

Column Descriptions and Schema Information:

- Drug: Name of Drug *<class 'str'>*
- STARTDATE: Date when the prescription started. *<class 'datetime'>*

Example 1: Analyze the clinical note to extract {{**Ibuprofen**}} data and document the findings in the EHR table. When Extract data, do not make assumptions or infer details not directly stated in the clinical note, even if this is common knowledge.
Clincal Note:
"1. Lisinopril 10 mg p.o. 7 days
2. {{**Ibuprofen **}} 400 mg Tablet Sig: One (1) Tablet PO Q8H (every 8 hours)"

Step 1) Identify and Extract Information about '{{**Acetaminophen**}}' in the given text.
[Answer in step 1]: DISCHARGE MEDICATIONS: ... {{**Ibuprofen **}} 400 mg Tablet

Step 2) Determine the STARTDATE and ENDDATE step-by-step: Identify when the '{{**Ibuprofen**}}' occurs.
[Answer in step 2]: When I reviewed the notes, it was mentioned that the STARTDATE would be the NaN. ENDDATE would be NAN.

Step 3) Based on the mentions of '{{**Ibuprofen**}}' in the given text found in Step 1 and Step 2, fill in the EHR table with these column headers: DRUG, STARTDATE, ENDDATE, DOSE_VAL_RX, DOSE_UNIT_RX.
[Answer in step 3]: As referred to in the answer from step 2, the drug is Ibuprofen. The starttime and enddate would be 'NAN'. The valuenum should be 400, and the dose_val_rx should be in mg.

Your Task: Analyze the clinical note to extract {{<<<<ENTITY>>>>}} data and document the findings in the EHR table. When Extract data, do not make assumptions or infer details not directly stated in the clinical note, even if this is common knowledge.

Clinical Note: "<<<<CLINICAL_NOTE>>>>"

Figure 17: Prompt Template for Pseudo Table Creation. The example is a prescriptions table.

---

**Prompt Template for Self Correction**

Task: You will be given a passage of clinical note along with several questions that relate to specific details within that clinical note. Your job is to determine whether the clinical note explicitly mentions the details asked in the questions.

For each question, your response should be divided into two parts:

- Evidence quote: Provide a direct quote or the exact sentences from the clinical note that either confirm or refute the detail in question. Additionally, include a brief explanation of why this evidence supports your answer.

- Answer: Respond with 'Yes' if the detail is explicitly mentioned in the clinical note using the exact words or phrases from the question. If the clnical note does not contain the specific detail, respond with 'No'. These are the only acceptable response options.

Example 1: Please answer the questions focusing on the specified entity named 'Hgb'.
Clinical Note:
"The patient admitted on 2196-2-18 and this note charted on 2196-2-21.
Pertinent Results: [**2200-07-01**] 09:25AM BLOOD WBC-7.7 RBC-3.16* {{**Hgb**}}-13.3* Hct-37.9* MCV-90 MCH-31.2 MCHC-32 RDW-12.7 Plt Ct-320"

Questions:

[1] Is it directly mentioned that Hgb's charttime is '2200-07-01'?
Evidence quote: "[**2200-07-01**] 09:25AM BLOOD WBC-7.7 RBC-3.16* {{**Hgb**}}-13.3* Hct-37.9*"
Answer: Yes. The clinical note explicitly mentions the date and time as "2200-07-01" in relation to the Hgb measurement, indicating the charttime for the Hgb value.

[2] Is it directly mentioned that Hgb's valuenum is '13.3'?
Evidence quote: "[**2200-07-01**] 09:25AM BLOOD WBC-7.7 RBC-3.16* {{**Hgb**}}-13.3* Hct-37.9*"
Answer: Yes. The note explicitly states the Hgb value as '13.3', directly mentioning the numerical value associated with the Hgb measurement.

[3] Is it directly mentioned that Hgb's valueuom is 'g/dL'?
Evidence quote: "[]"
Answer: No. The clinical note mentions "BLOOD Hgb-13.3" but does not specify the unit of measurement for the Hgb value.

Your task: Please answer the questions focusing on the specified entity named "<<<<EN-TITY>>>>".
Clinical Note:
"The patient admitted on <<<<ADMISSION>>>> and this note charted on <<<<CHART-TIME>>>>.
<<<<CLINICAL_NOTE>>>>"

Questions:
Please maintain the output format.
<<<<QUESTIONS>>>>

---

Figure 18: Prompt Template for Self Correction.

## Prompt Template for Value Reformatting

Task: Transform Given Data to Match a Database Table Format. Your goal is to modify a set of given data so that it matches the format of an existing database table. The data transformation should adhere to any constraints evident from the table's structure. Focus solely on the fields provided in the given data and do not add or infer any additional fields. Only transform and include the fields mentioned in the given data. Do not add, create, or infer any additional fields beyond what is specified.

Time Information:

- Existing Table Schema: {'Chartevents': {'CHARTTIME': *<class 'datetime'>*, 'VALUENUM': *<class 'float'>*, 'VALUEUOM' : *<class 'str'>*}, 'D_items': {'LABEL': *<class 'str'>*}}
- Example rows of Chartevents and D_items tables

    - Chartevents.CHARTTIME: 2200-11-03
    - Chartevents.VALUENUM: 30.0
    - Chartevents.VALUEUOM: %
    - D_items.LABEL: Monocytes

- CHARTTIME uses a 24-hour format.
- If the given data includes relative dates, replace it with the corresponding actual date from the patient's record.

Example 1)
- Information:

    - Admission Date: 1999-11-11
    - Date charted on the note: 1999-11-13
- Given Data:

    - Chartevents.CHARTTIME: [**2208-11-08**] 8:00 PM
    - Chartevents.VALUENUM: 7.10
    - Chartevents.VALUEUOM: mg/dL
    - D_items.LABEL: Blasts
- Output:

    - Chartevents.CHARTTIME: 2208-11-08 20:00:00
    - Chartevents.VALUENUM: 7.10
    - Chartevents.VALUEUOM: mg/dL
    - D_items.LABEL: Blasts

Your Task:
- Information

    - Admission Date: <<<ADMISSION>>>
    - Date charted on the note: <<<CHARTTIME>>>
- Given Data: <<<GIVEN_DATA>>>
- Output: [Only print answer here with structured format tablename.columnname = value]

Figure 19: Prompt Template for Value Reformatting. The example is a Chartevents table.

---

**Prompt Template for Query Generation**

Task: You are a highly intelligent and accurate sqlite3 query creator. You take a [{table-name}.{columnname} = {condition value}] and given extra information and turn it into a [SQLite3 query]. Please Use only the information given. Your output format is a dictionary with a single key 'Q' and the value is the SQLite3 query, so [{'Q': Query}] form. And begin the query with "SELECT *", to retrieve all columns.

Rules:
- Utilize strftime to maintain the given time format.
- The 'Chartevents' and 'D_items' tables need to be joined using the 'itemid' as the key for the join operation.

Example 1)
[{table}.{column} = {condition value}]
Chartevents.hadm_id = 12345
Chartevents.valuenum = 94.0
Chartevents.valueuom = mmHg
Chartevents.charttime = '2000-11-11'
D_items.label = 'BP'

Output: [{"Q": "SELECT * FROM Chartevents JOIN D_items ON Chartevents.itemid = D_items.itemid WHERE Chartevents.hadm_id = 12345 AND Chartevents.valuenum = 94.0 AND Chartevents.valueuom = 'mmHg' AND strftime('%Y-%m-%d', Chartevents.charttime) = '2000-11-11' AND D_items.label = 'BP'"}]

Example 2)
[{table}.{column} = {condition value}]
Chartevents.hadm_id = 14456
Chartevents.valuenum = 36.7
D_items.label = 'Temp'
Chartevents.charttime = '2331-02-11 21:32:33'

Output:[{"Q": "SELECT * FROM Chartevents JOIN D_items ON Chartevents.itemid = D_items.itemid WHERE Chartevents.hadm_id = 14456 AND Chartevents.valuenum = 36.7 AND D_items.label = 'Temp' AND strftime('%Y-%m-%d %H:%M:%S', Chartevents.charttime) = '2331-02-11 21:32:33'"}]

Your task)
[{table}.{column} = {condition value}]
Chartevents.hadm_id = <<<<HADM_ID>>>>

Output: [Write your answer here]

---

Figure 20: Prompt Template for Query Generation. This is an example where the date is written in the yyyy-mm-dd (hh:mm:ss) format.

# H Note Segmentation

Algorithm 1 provides a summary of the note segmentation process, with a detailed example illustrated in Figure 21.

---

**Algorithm 1** Note Segmentation Process

---

**Require:** Clinical Note $P$, Maximum Length of Subtext $l$, Number of Subtexts $n$
**Ensure:** Set of Subtexts $\mathcal{T}$
1: $\mathcal{T} \leftarrow \emptyset, i \leftarrow 0, P_i \leftarrow P$
2: **while** $TokenLen(P_i) > l$ **do**
3: $\quad P_i^f, P_i^b \leftarrow DivideByLen(P_i, l)$
4: $\quad P_{i,1}^f, P_{i,2}^f, ..., P_{i,n}^f, \leftarrow DivideByLLM(P_i^f, n)$
5: $\quad \mathcal{T} \leftarrow \mathcal{T} \cup \{P_{i,1}^f, P_{i,2}^f, ...P_{i,n-1}^f\}$
6: $\quad P_{i+1} \leftarrow MergeText(P_{i,n}^f, P_i^b)$
7: $\quad i \leftarrow i + 1$
8: **end while**
9: $\mathcal{T} \leftarrow \mathcal{T} \cup \{P_i\}$
10: $\mathcal{T} \leftarrow MakeIntersection(\mathcal{T})$
11: **return** $\mathcal{T}$

---

Figure 21: Overall process of Note Segmentation where $n$ is 3.

## H.1 Effectiveness of Note Segmentation

Recent advancements in LLMs with extended context lengths have introduced the possibility of processing entire clinical notes without segmentation, potentially capturing more contextual information for error detection tasks. However, upon evaluating this approach on 25% of our test set, we found that models processing unsegmented notes consistently showed a significant decrease in recall compared to their segmented counterparts—as seen in Table 7, while there was a slight increase in precision for some models, the overall ability to detect errors diminished. GPT-3.5 with a 16k context window (without segmentation) achieved a recall of 48.64% and precision of 54.53%, whereas with segmentation (using the 4k context window), it achieved a higher recall of 65.45% and a precision of 50.75%. This suggests that current long-context LLMs may struggle with accurately extracting detailed information from extensive free-form clinical notes, leading to missed errors. Therefore, our note segmentation approach remains effective, as it enhances the models' ability to detect errors by focusing on relevant sections and allows the use of models with shorter context lengths, which are more accessible and practical for deployment across different healthcare settings.

Table 7: Experiment Results for Note Segmentation

| Experiment Setting | Model | Recall | Precision |
|---|---|---|---|
| w/ Note Segmentation | GPT-3.5 4k (0613) | 65.45 | 50.75 |
| | Llama 3 70B | 53.86 | 47.01 |
| | Mixtral 8X7B | 54.92 | 44.60 |
| w/o Note Segmentation | GPT-3.5 16k (0613) | 48.64 | 54.53 |
| | Llama 3.1 70B | 35.23 | 45.40 |
| | Mixtral 8X7B | 36.43 | 42.58 |

# I   Process of Creating Pseudo Table

An example of the process for creating a pseudo table is described in Figure 22.

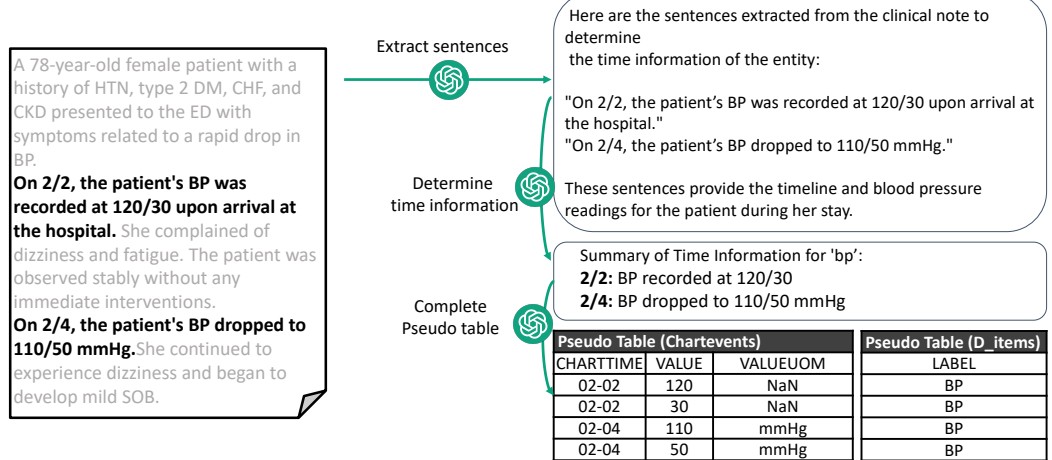

Figure 22: The overall process of creating a pseudo table. First, the LLM identifies sentences about a given entity (*BP*) in the text. Then, it finds time information in those sentences and uses this information to complete the pseudo table.

# J   Hallucination of LLMs

Examples of hallucination in LLMs are described in Figure 23.

# K   Experiments using Original Notes

The results using the unfiltered original notes are reported in Table 8. Clinical notes often contain not only information related to the current hospitalization but also past medical history or future plans, which can be difficult to discern from tables. To effectively filter this information, we use a Time Filtering step where an LLM determines if the entity occurred during the current hospitalization. We proceed with further steps (*e.g.*, pseudo table creation) only if the LLM determines that the entity is relevant to the current hospitalization. The experimental results showed that Recall and Precision decreased by approximately 12.73% and 8.31%, respectively, compared to the filtered notes. This indicates that the current LLMs lack the reasoning ability to understand clinical notes and determine whether each event occurred during the current admission.

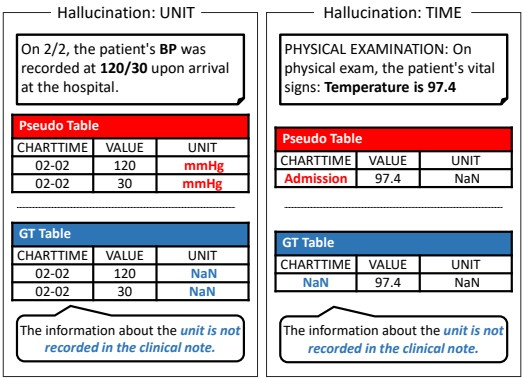

Figure 23: Examples of hallucination in LLMs. There have been many instances where the LLM generated non-existent information from clinical notes to create pseudo tables. In the left box, a pseudo table was created with the unit listed as 'mmHg', which was not mentioned in the note. In the right box, a pseudo table was created with the charttime listed as 'admission', based on the 'PHYSICAL EXAMINATION' section, which was not explicitly mentioned.

Table 8: Results using the unfiltered original notes. We conduct experiments using a few-shot setting.

| Models | Discharge Summary | | | Physician Note | | | Nursing Note | | | Total | | |
|---|---|---|---|---|---|---|---|---|---|---|---|---|
| | Rec | Prec | Inters | Rec | Prec | Inters | Rec | Prec | Inters | Rec | Prec | Inters |
| Tulu2 | 31.90 | 40.75 | 69.18 | 39.78 | 43.43 | 85.64 | 39.07 | 29.10 | 68.56 | 36.91 | 37.76 | 74.46 |
| Mixtral | 39.16 | 37.32 | 72.52 | 37.55 | 40.86 | 99.11 | 47.86 | 31.38 | 70.08 | 41.52 | 36.52 | 81.57 |
| Llama-3 | 41.91 | 36.88 | 69.34 | 37.2 | 40.44 | 79.37 | 32.09 | 27.20 | 54.23 | 37.06 | 34.84 | 67.64 |

# L Experiments Setting Details

## L.1 Examples of Evaluation Metrics

We measured the performance of the framework using the following three metrics: Recall, Precision, and Intersection. We demonstrate how these metrics are calculated with the following examples. Consider the gold entity set $\mathcal{G} = \{\{e_1, i\}, \{e_2, c\}, \{e_3, c\}\}$, and the recognized entity set $\mathcal{R} = \{\{e_1, i\}, \{e_3, i\}, \{e_4, c\}\}$, where $e_n$ represents an entity, $i$ indicates INCONSISTENT, and $c$ indicates CONSISTENT. In this situation, the Recall is 33.33% because only 1 out of the 3 labeled entities, $\{e_1, i\}$, was correctly classified. Similarly, the Precision is 33.33% since only 1 out of the 3 recognized entities, $\{e_1, i\}$, was accurate. However, the Intersection is 50.00% because the intersection of the gold and recognized sets includes 2 entities, $\{e_1, i\}$ and $\{e_3, i\}$, and only 1 of these, $\{e_1, i\}$, was correctly classified.

## L.2 Number of In-Context Examples

We carefully designed in-context examples for each stage to maximize the performance of the framework. We provided 15 examples for Table Identification stage and 2 examples for all other stages.

# M Results of MIMIC-III and MIMIC-OMOP

Table 9 presents the experimental results categorized by entity type and note type from both MIMIC-III and MIMIC-OMOP.

Table 9: The results of MIMIC-III and MIMIC-OMOP. This performance metric is Recall.

| Data | Type | Model | Discharge Summary | Physician Note | Nursing Note | Total |
|---|---|---|---|---|---|---|
| **MIMIC-III** | **Type 1** | Tulu2 | 69.26 | 73.38 | 68.75 | 66.79 |
| | | Mixtral | 68.70 | 75.50 | 68.93 | 59.21 |
| | | Llama3 | 65.15 | 72.88 | 59.21 | 65.74 |
| | **Type 2** | Tulu2 | 19.56 | 43.06 | 40.34 | 34.32 |
| | | Mixtral | 19.49 | 47.38 | 49.47 | 38.31 |
| | | Llama3 | 20.10 | 42.76 | 56.68 | 39.84 |
| **MIMIC-OMOP** | **Type 1** | Tulu2 | 54.93 | 52.41 | 54.75 | 54.03 |
| | | Mixtral | 51.88 | 57.34 | 47.98 | 52.40 |
| | | Llama3 | 51.17 | 53.35 | 53.08 | 52.53 |
| | **Type 2** | Tulu2 | 49.77 | 47.13 | 47.05 | 47.98 |
| | | Mixtral | 50.50 | 35.41 | 50.41 | 45.44 |
| | | Llama3 | 54.16 | 53.23 | 49.85 | 52.41 |

# N Component Analysis of CheckEHR

All main experiments utilized the Item Search Tool (see Sec. 3.2) to search for items related to entities in the database. However, the actual annotated data also includes instances where annotators manually searched for items in the database when the search tool did not yield results. As seen in Table 11, the experiment showed that using both the outputs from the Item Search Tool and the additional manual annotations by annotators resulted in an increase in Recall and Precision by 1.75% and 4.85%, respectively, compared to using only the Item Search Tool. Therefore, future work should explore methods to find semantically similar items in the database, rather than relying solely on the surface form of the entity.

Table 10 shows the results of experiments for component analysis conducted by matching entities with items in the database, including items added by the Item Search Tool and annotators. The open-source models also demonstrated an average performance improvement of 10% in each setting, proving that each stage plays a crucial role in solving this task. Unlike other models that showed significant performance improvements in the second and third settings, Llama3's Recall remained at 74.96% in the third setting. This indicates that Llama3's SQL query generation capability is inferior to that of other models. By understanding the performance of each LLM in different settings and addressing their shortcomings, the framework's performance will be significantly enhanced.

Table 10: Component Analysis Result

| Models | Experiment Setting | Rec | Prec | Models | Experiment Setting | Rec | Prec |
|---|---|---|---|---|---|---|---|
| **Tulu2** | CheckEHR | 50.69 | 45.93 | **Llama3** | CheckEHR | 53.86 | 41.57 |
| | - NER | 61.86 | 74.27 | | - NER | 60.33 | 64.67 |
| | - (Table Identification + Time Filtering) | 67.53 | 69.69 | | - (Table Identification + Time Filtering) | 67.12 | 68.81 |
| | - Pseudo Table Creation | 90.35 | 93.27 | | - Pseudo Table Creation | 74.96 | 87.32 |
| **Mixtral** | CheckEHR | 54.92 | 44.60 | **GPT-3.5 (0613)** | CheckEHR | 65.45 | 50.75 |
| | - NER | 66.35 | 70.51 | | - NER | 76.11 | 77.70 |
| | - (Table Identification + Time Filtering) | 70.05 | 74.86 | | - (Table Identification + Time Filtering) | 82.49 | 78.99 |
| | - Pseudo Table Creation | 86.06 | 91.26 | | - Pseudo Table Creation | 92.83 | 94.93 |

Table 11: Results of experiments for component analysis conducted by matching entities with items in the database. In our experiment, we bypassed the named entity recognition (NER) step and directly provided gold entities. -Additional Manual Annotation uses only the Item Search Tool, whereas +Additional Manual Annotation uses both the Item Search Tool and additional manual annotation.

| Model | -Additional Manual Annotation | | +Additional Manual Annotation | |
|---|---|---|---|---|
| | Rec | Prec | Rec | Prec |
| Tulu2 | 61.56 | 65.07 | 61.86 | 74.27 |
| Mixtral | 65.07 | 70.21 | 66.35 | 70.51 |
| Llama3 | 59.23 | 56.16 | 60.33 | 64.67 |
| GPT-3.5 (0613) | 71.81 | 76.25 | 76.11 | 77.70 |

## O    Sample data of `EHRCon`

The sample data of `EHRCon` is described in Figure 24.

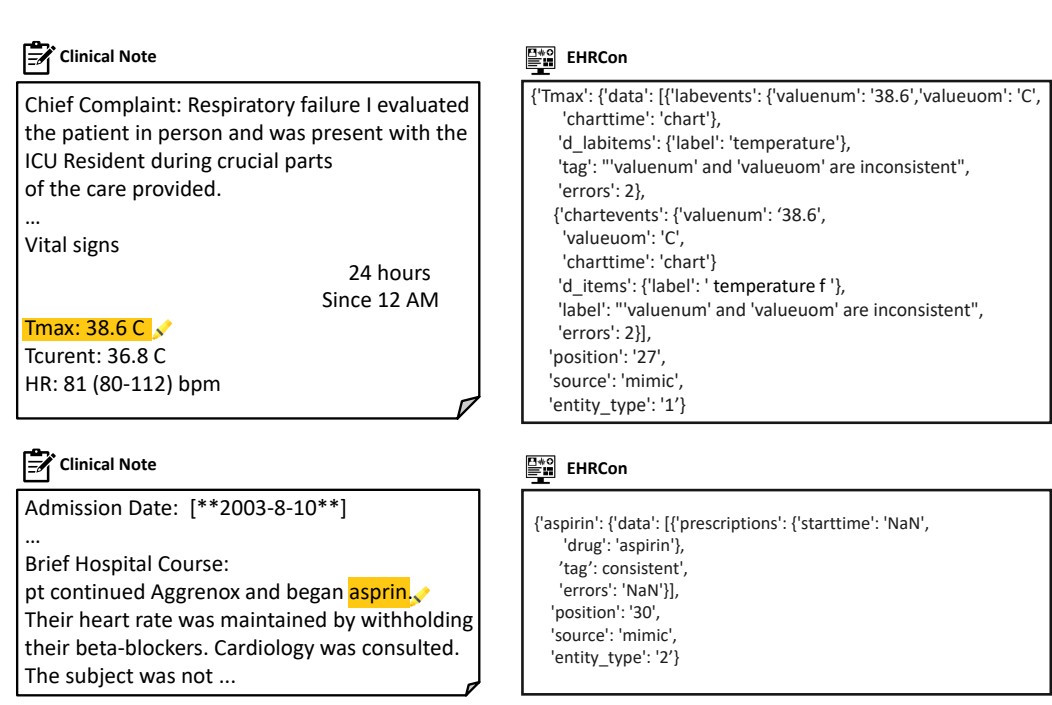

Figure 24: These are sample datas of `EHRCon`. The 'Tmax' example illustrates an inconsistent case, while the 'aspirin' example demonstrates a consistent case. The 'tag' provides the results of the verification, the 'errors' indicate the number of errors, and the 'position' refers to the sentence number containing each entity. The 'entity_type' of '1' indicates that a numeric value is clearly shown, while the 'entity_type' of '2' means that the presence of the entity in the database is sufficient for verification. If the 'time' is marked as 'NaN', it means there is no time information associated with the entity. In the provided samples, all entities and values have been altered to prevent patient identification, and the sentences within the clinical notes have been rephrased with some content added or removed.

# P  SQL Generation Method

In our data annotation process, we adopted a template-based method for generating SQL queries to ensure both consistency and precision. Annotators filled in predefined SQL templates, with specific slots for table names, column names, patient identifiers, time expressions, and condition values *(e.g., SELECT * FROM* {table1} *JOIN* {table2} *ON* {table1}.{column_name1} = {table2}.{column_name2} *WHERE* {table1}.hadm_id = {hadm_id} *AND* [time_value_template] *AND* {table1}.[condition_value_column] = {condition_value}).

To cover a range of time expressions found in clinical notes, the [time_value_template] accommodated not only standard formats such as '*YYYY-MM-DD HH:MM:SS*' but also narrative descriptors like '*admission*' or '*discharge*'. It even handled cases where time information was entirely absent, allowing for flexibility in the SQL query generation process. As a result, 27 unique templates were developed, all recorded in Table 12, and carefully reviewed and validated by two authors. This validation process included simulating potential queries and cross-checking the results against the underlying database to ensure the templates accurately captured the intended data retrieval logic.

Annotators relied on these validated templates to generate SQL queries throughout the annotation task, which significantly contributed to maintaining high reliability in data extraction. This approach's effectiveness was further supported by a high inter-annotator agreement rate of 93.8%, as seen in cross-checked annotations (refer to Appendix E). This consistency in the annotations indirectly validated the accuracy of the generated SQL queries, reinforcing the robustness of our approach.

Table 12: SQL template used for data annotation

| Table | SQL template |
|---|---|
| Chartevents | SELECT * FROM Chartevents JOIN d_items ON Chartevents.itemid d_items.itemid WHERE strftime('%Y-%m-%d', Chartevents.charttime) = '{date_value}' AND {condition_values}; |
| | SELECT *FROM Chartevents JOIN d_items ON Chartevents.itemid = d_items.itemid WHERE strftime('%Y-%m-%d', Chartevents.charttime) BETWEEN strftime('%Y-%m-%d', datetime('{admission}', '-1 day')) AND strftime('%Y-%m-%d', datetime('{admission}', '+1 day'))AND {condition_value}; |
| | SELECT * FROM Chartevents JOIN d_items ON Chartevents.itemid = d_items.itemid WHERE strftime('%Y-%m-%d', Chartevents.charttime) BETWEEN strftime('%Y-%m-%d', datetime('{admission}', '-1 day')) AND strftime('%Y-%m-%d', datetime('{chart}', '+1 day')) AND {condition_value}; |
| | SELECT * FROM Chartevents JOIN d_items ON Chartevents.itemid = d_items.itemid WHERE strftime('%Y-%m-%d', Chartevents.charttime) BETWEEN strftime('%Y-%m-%d', datetime('{calculated_time}', '-1 day')) AND strftime('%Y-%m-%d', datetime('{calculated_time}', '+1 day')) AND {condition_value}; |
| Labevents | SELECT * FROM Labevents JOIN d_labitems ON Labevents.itemid = d_labitems.itemid WHERE strftime('%Y-%m-%d', Labevents.charttime) = '{date_value}' AND {condition_values}; |
| | SELECT * FROM Labevents JOIN d_labitems ON Labevents.itemid = d_labitems.itemid WHERE strftime('%Y-%m-%d',chartevents.charttime) BETWEEN strftime('%Y-%m-%d', datetime('admission', '-1 day')) AND strftime('%Y-%m-%d', datetime('admission', '+1 day'))AND {condition_value}; |
| | SELECT * FROM Labevents JOIN d_labitems ON Labevents.itemid = d_labitems.itemid WHERE strftime('%Y-%m-%d', Labevents.charttime) BETWEEN strftime('%Y-%m-%d', datetime('{admission}', '-1 day')) AND strftime('%Y-%m-%d', datetime('{chart}', '+1 day')) AND {condition_value}; |
| | SELECT * FROM Labevents JOIN d_labitems ON Labevents.itemid = d_labitems.itemid WHERE strftime('%Y-%m-%d', chartevents.charttime) BETWEEN strftime('%Y-%m-%d', datetime('{calculated_time}', '-1 day')) AND strftime('%Y-%m-%d', datetime('{calculated_time}', '+1 day')) AND {condition_value}; |
| Inputevents_cv | SELECT * FROM Inputevents_cv JOIN d_items ON Inputevents_cv.itemid = d_items.itemid WHERE strftime('%Y-%m-%d', Inputevents_cv.charttime) ='{date_value}' AND {condition_values}; |
| | SELECT * FROM Inputevents_cv JOIN d_items ON Inputevents_cv.itemid = d_items.itemid WHERE strftime('%Y-%m-%d',Inputevents_cv.charttime) BETWEEN strftime('%Y-%m-%d',datetime('{admission}', '-1 day')) AND strftime('%Y-%m-%d',datetime('{admission}', '+1 day')) AND {condition_value}; |
| | SELECT * FROM Inputevents_cv JOIN d_items ON Inputevents_cv.itemid = d_items.itemid WHERE strftime('%Y-%m-%d',Inputevents_cv.charttime) BETWEEN strftime('%Y-%m-%d', datetime('{admission}', '-1 day')) AND strftime('%Y-%m-%d',datetime('{chart}', '+1 day')) AND {condition_value}; |
| Inputevents_mv | SELECT * FROM Inputevents_mv JOIN d_items ON Inputevents_mv.itemid = d_items.itemid WHERE strftime('%Y-%m-%d','{date_value}') BETWEEN strftime('%Y-%m-%d', Inputevents_mv.starttime) AND strftime('%Y-%m-%d', Inputevents_mv.endtime) AND {condition_value}; |
| | SELECT * FROM Inputevents_mv JOIN d_items ON Inputevents_mv.itemid = d_items.itemid WHERE NOT (strftime('%Y-%m-%d', Inputevent_mv.endtime) < datetime('{admission}', '-1 day') OR strftime('%Y-%m-%d', Inputevents_mv.starttime) > datetime ('{admission}', '+1 day')) AND {condition_value}; |
| | SELECT * FROM Inputevents_mv JOIN d_items ON Inputevents_mv.itemid = d_items.itemid WHERE NOT (strftime('%Y-%m-%d',Inputevents_mv.endtime) < datetime('{admission}', '-1 day') OR strftime('%Y-%m-%d', Inputevents_mv.starttime) > datetime ('{chart}', '+1 day'))AND {condition_value}; |

Table 12: SQL template used for data annotation (Continued)

| Table | SQL template |
|---|---|
| | SELECT * FROM Inputevents_mv JOIN d_items ON Inputevents_mv.itemid = d_items.itemid WHERE NOT (strftime('%Y-%m-%d',Inputevent_mv.endtime) < datetime('{calculated_time}', '-1 day') OR strftime('%Y-%m-%d', Inputevents_mv.starttime) > datetime ('{calculated_time}', '+1 day'))AND {condition_value}; |
| Microbiologyevents | SELECT * FROM Microbiologyevents JOIN D_items AS spec_items ON Microbiologyevents.spec_itemid = spec_items.itemid JOIN D_items AS org_items ON Microbiologyevents.org_itemid = org_items.itemid WHERE strftime('%Y-%m-%d', Microbiologyevents.charttime) = '{date_value}' AND {condition_value}; |
| | SELECT * FROM Microbiologyevents JOIN D_items AS spec_items ON Microbiologyevents.spec_itemid = spec_items.itemid JOIN D_items AS org_items ON Microbiologyevents.org_itemid = org_items.itemid WHERE strftime('%Y-%m-%d', microbiologyevents.charttime) BETWEEN strftime('%Y-%m-%d', datetime('{admission}', '-1 day')) AND strftime('%Y-%m-%d', datetime('{admission}', '+1 day'))AND {condition_value}; |
| | SELECT *FROM Microbiologyevents JOIN D_items AS spec_items ON Microbiologyevents.spec_itemid = spec_items.itemid JOIN D_items AS org_items ON Microbiologyevents.org_itemid = org_items.itemid WHERE strftime('%Y-%m-%d', microbiologyevents.charttime) BETWEEN strftime('%Y-%m-%d', datetime('{admission}', '-1 day')) AND strftime('%Y-%m-%d', datetime ('{chart}', '+1 day')) AND {condition_value}; |
| Prescriptions | SELECT * FROM Prescriptions WHERE strftime('%Y-%m-%d', '{date_value}') BETWEEN strftime('%Y-%m-%d', Prescriptions.startdate) AND strftime('%Y-%m-%d', Prescriptions.enddate) AND {condition_value}; |
| | SELECT * FROM Prescriptions WHERE NOT (strftime('%Y-%m-%d', Prescriptions.enddate) < datetime('{admission}', '-1 day') OR strftime('%Y-%m-%d', Prescriptions.startdate) > datetime('{admission}', '+1 day')) AND {condition_value}; |
| | SELECT * FROM Prescriptions WHERE NOT (strftime('%Y-%m-%d', Prescriptions.enddate) < datetime('{admission}', '-1 day') OR strftime('%Y-%m-%d', Prescriptions.startdate) > datetime('{chart}', '+1 day')) AND {condition_value}; |
| | SELECT * FROM Prescriptions WHERE NOT (strftime('%Y-%m-%d', Prescriptions.enddate) < datetime('{calculated_time}', '-1 day') OR strftime('%Y-%m-%d', Prescriptions.startdate) > datetime('{calculated_time}', '+1 day')) AND {condition_value}; |
| Outputevents | SELECT * FROM Outputevents JOIN d_items ON Outputevents.itemid = d_items.itemid WHERE strftime('%Y-%m-%d', Outputevents.charttime) = '{date_value}' AND {condition_values}; |
| | SELECT * FROM Outputevents JOIN d_items ON Outputevents.itemid = d_items.itemid WHERE strftime('%Y-%m-%d',Outputevents.charttime) BETWEEN strftime('%Y-%m-%d', datetime('{admission}', '-1 day')) AND strftime('%Y-%m-%d', datetime('{admission}', '+1 day'))AND {condition_value}; |
| | SELECT * FROM Outputevents JOIN d_items ON Outputevents.itemid = d_items.itemid WHERE strftime('%Y-%m-%d', Outputevents.charttime) BETWEEN strftime('%Y-%m-%d',datetime('{admission}', '-1 day')) AND strftime('%Y-%m-%d', datetime('{chart}', '+1 day'))AND {condition_value}; |
| Procedures_icd | SELECT * FROM Procedures_icd JOIN D_icd_procedures ON Procedures_icd.ICD9_CODE=D_icd_procedures.ICD9_CODE WHERE Procedures_icd.hadm_id=hadm_idAND (D_icd_procedures.LONG_TITLE='{long_title}' OR D_icd_procedures.SHORT_TITLE='{short_title}') |
| Diagnoses_icd | SELECT * FROM Diagnoses_icd JOIN d_icd_diagnoses ON Diagnoses_icd.ICD9_CODE=d_icd_diagnoses.ICD9_CODE WHERE Diagnoses_icd.hadm_id=hadm_id AND (d_icd_diagnoses.LONG_TITLE='{long_title}' OR d_icd_diagnoses.SHORT_TITLE='{short_title}') |

## Q   Author statement

The authors of this paper bear all responsibility in case of violation of rights, etc. associated with `EHRCon`.