# OpenReview forum: "EHRCon: Dataset for Checking Consistency between Unstructured Notes and Structured Tables in Electronic Health Records"
_NeurIPS.cc/2024/Datasets_and_Benchmarks_Track — NeurIPS 2024 Track Datasets and Benchmarks Spotlight_

### Official Review · Reviewer_9y8Y · 2024-07-24
**Review comments**

**Rating:** 8
**Confidence:** 4
**Correctness:** The claims are correct.
**Clarity:** The paper is well written.

**Review:**

I have a few minor comments:

1. Is the type of inconsistency recorded in the labels? For example, are some records noted as missing, while others may record incorrect values?
2. What are the differences between inconsistencies found in notes versus those found in tables? How do you determine which one is correct?
3. An analysis of the incorrect predictions made by the proposed model could be helpful to understand which types of errors current models are incapable of addressing.

**Strengths:**

The topic is novel and clinically significant. Generally, this paper is well-suited for a benchmark track.

**Additional Feedback:**

NA

**Documentation:**

NA

**Limitations:**

Please refer to my review comments.

**Opportunities For Improvement:**

Please refer to my review comments.

**Relation To Prior Work:**

Related works are discussed.

**Summary And Contributions:**

In this work, the authors propose an EHR data consistency dataset using the MIMIC-III dataset. The dataset is human-labeled to identify inconsistencies between EHR tables and clinical notes. Additionally, a LLM-based framework is proposed to address these consistency issues.

---

> ### Author Rebuttal · Authors · 2024-08-17
>
> We are grateful for your valuable feedback and suggestions. We address your comments by following Q-A pairs (from **Q1** to **Q3**).
>
> **Q1. Is the type of inconsistency recorded in the labels? For example, are some records noted as missing, while others may record incorrect values?**
>
> A1: As mentioned in Line 148 of the paper, we have tagged the types of inconsistencies within the data, such as time inconsistencies.
>
>
> **Q2. What are the differences between inconsistencies found in notes versus those found in tables? How do you determine which one is correct?**
>
> A2: As mentioned in Lines 97-104, we acknowledge that both types of data—clinical notes and structured data—can contain errors. Therefore, rather than attributing inconsistencies to a specific source, we have taken a more flexible approach by defining labels as Consistent / Inconsistent, without relying on a specific assumption.
>
>
> **Q3. An analysis of the incorrect predictions made by the proposed model could be helpful to understand which types of errors current models are incapable of addressing.**
>
> A3: We conducted a detailed component analysis as outlined in Section 5.4 to gain deeper insights into the types of errors that current models struggle to address. This analysis was performed through three distinct experimental settings. In the first setting, we removed the NER stage and provided the models with ground truth entities. Building upon this, the second setting included ground truth for both the time filtering and table identification stages. In the third setting, we further incorporated ground truth for pseudo table creation, self-correction, and value reformatting stages. As we progressed through these stages, we observed average performance improvements of 9.93, 7.25, and 14.25, respectively. This result clearly indicates that current models face significant challenges, particularly in mapping long free-form text to structured tables.

---

> ### Comment · Reviewer_9y8Y · 2024-08-19
> **Response to author rebuttal**
>
> Thanks to the authors for their response. My questions have been addressed. I believe this is a good paper in the benchmark track.

---

> > ### Author Rebuttal · Authors · 2024-08-19
> >
> > Thank you very much for your positive feedback on our work. If you have any further questions, please feel free to reach out to us anytime.

---

### Official Review · Reviewer_b7rr · 2024-07-24
**Intersting and Impactful Work**

**Rating:** 8
**Confidence:** 4
**Correctness:** Yes
**Clarity:** Yes

**Review:**

The paper is of the highest standard in terms of quality, clarity, originality, and significance. The EHRCon dataset has the potential to facilitate the development of foundational EHR models that heavily rely on paired (consistent) clinical notes and structured EHR data. It is a significant contribution to the field and should definitely be accepted.

**Strengths:**

- High Quality: The paper provides a thorough and rigorous evaluation of the dataset and framework.
- Clarity: The writing is clear, well-structured, and supplemented by helpful figures, tables, and supplementary materials.
- Originality: The dataset and framework fill a significant gap in current research by addressing the consistency between unstructured and structured EHR data.
- Significance: The work is highly relevant and impactful for the healthcare AI research community, with direct applications in EHR foundation models.

**Additional Feedback:**

Good paper

**Documentation:**

Yes

**Limitations:**

No obvious limitations were found in the work.

**Opportunities For Improvement:**

While the paper is strong overall, a few areas for improvement include:

- Broader Model Evaluation: Evaluate more LLMs to provide a more comprehensive assessment.
- Statistical Reporting: Include mean and standard deviation scores in the main results to improve robustness.
- Dataset Comparison: Evaluate and compare the framework using MIMIC-IV and other datasets to demonstrate broader applicability.

**Relation To Prior Work:**

Yes

**Summary And Contributions:**

The paper introduces EHRCon, a dataset designed to ensure the consistency between unstructured clinical notes and structured tables in EHR data. The primary contributions include:

- EHRCon Dataset: Annotated dataset with 3,943 entities across 105 clinical notes checked against database entries for consistency.
- CheckEHR Framework: An eight-stage process utilizing large language models (LLMs) to verify consistency between clinical notes and database entries.
- Evaluation: Comprehensive evaluation using the MIMIC-III dataset and its OMOP CDM schema version, showcasing the framework's performance in both few-shot and zero-shot settings.

---

> ### Author Rebuttal · Authors · 2024-08-17
>
> We are grateful for your valuable feedback and suggestions. We address your comments by following Q-A pairs (from **Q1** to **Q3**).
>
> **Q1. Broader Model Evaluation: Evaluate more LLMs to provide a more comprehensive assessment.**
>
> A1: We appreciate your suggestion and conducted additional experiments using the recently released models, including Qwen-2 72B, and Gemma-2 27B, incorporating these results into the final version of the paper.
> Due to time constraint, we were only able to run these additional experiments on 25% of the test set. The results for the Gemma-2 27B model are complete, and unfortunately, it scored 0, suggesting that a model with at least 50B parameters is necessary to effectively perform a consistency check between clinical notes and EHR tables. As for the Qwen-2 72B model, the experiment is still in progress, but we are working diligently to complete it and will report the results within the discussion period.
> Additionally, we are committed to running experiments on the entire test set with both Qwen-2 72B and Gemma-2 27B and will ensure that these comprehensive results are reflected in the final version of the paper.
>
> **Q2. Statistical Reporting: Include mean and standard deviation scores in the main results to improve robustness.**
>
> A2: We plan to conduct additional experiments to include confidence intervals. While it may be challenging to complete all experiments by the rebuttal deadline, we aim to report these results before the end of the discussion period. If we are unable to complete the experiments in time, we will ensure they are included in the final version.
>
>
> **Q3. Dataset Comparison: Evaluate and compare the framework using MIMIC-IV and other datasets to demonstrate broader applicability.**
>
> A3: Among publicly available EHR databases, MIMIC is the only one that provides paired notes and tables. However, as mentioned in the footnote on page 3, we opted to use the MIMIC-III dataset instead of MIMIC-IV. Although MIMIC-IV is a more recent dataset, it lacks diverse note types (such as physician and nursing notes) and has all dates removed from notes for de-identification, rather than shifting the dates as done in MIMIC-III. This made MIMIC-IV less suitable for our analysis.

---

> > ### Comment · Reviewer_b7rr · 2024-08-19
> >
> > Thanks for your response and ongoing experiments! Very nice paper; I hope you receive the oral or spotlight presentation!
> >
> > By the way, could you release the curated MIMIC dataset (with paired notes and tables, or single modality with an indication of missing modality for the other)? I would be very happy to use the dataset for my future work. In addition, I believe that the curated dataset could serve as a reflection on the existing multimodal EHR-related papers, since previous work has not checked the consistency but simply fused multiple modalities.

---

> > > ### Author Rebuttal · Authors · 2024-08-19
> > >
> > > Thank you very much for your positive feedback on our work.
> > >
> > > We have submitted EHRCon to Physionet, and it is currently under review. Once the review process is completed, EHRCon will be available for download on Physionet.
> > >
> > > In response to your suggestion that we experiment with more models, we have conducted experiments using the Gemma-2 27B and Qwen-2 72B models on 25% of the entire test set. The results are summarized in the table below. Notably, Qwen-2 achieved the highest precision among all the models we have tested so far. We appreciate your valuable feedback, as it has helped us further improve our research.
> > >
> > > If you have any further questions, we would be happy to address them.
> > >
> > > --------
> > > | Model            | Recall | Precision |
> > > |-------------------|---------|-----------|
> > > | Gemma-2 27B  | 0.00 | 0.00   |
> > > | Qwen-2 72B   | 57.35 | 53.46 |

---

> > > > ### Author Rebuttal · Authors · 2024-08-19
> > > >
> > > > We would like to inform you that the EHRCon dataset has been accepted by PhysioNet, and the citation information has been updated. Once the final copyediting and approval process is complete, the dataset will be publicly available.
> > > >
> > > > The updated citation information is as follows:
> > > > *Kwon, Y., Kim, J., Lee, G., Bae, S., Kyung, D., Cha, W., Pollard, T., Johnson, A., & Choi, E. (2024). EHRCon: Dataset for Checking Consistency between Unstructured Notes and Structured Tables in Electronic Health Records (version 1.0.0). PhysioNet. https://doi.org/10.13026/x1ea-np24*
> > > >
> > > > Please note that the DOI will be activated once the dataset is officially published.

---

> ### Comment · Reviewer_b7rr · 2024-08-19
>
> That's great, congratulations!

---

### Official Review · Reviewer_V5Jt · 2024-07-25
**Dataset for Checking Consistency between Unstructured Notes and EHR tables.**

**Rating:** 8
**Confidence:** 3
**Correctness:** The analysis appears to be sound.
**Clarity:** The paper is very well written.

**Review:**

This work addresses an important problem in healthcare informatics - inconsistencies in EHRs that can potentially impact patient safety. It provides both a benchmark dataset for future research and a novel approach to automating consistency checks in clinical documentation. The paper demonstrates high-quality research with a well-designed dataset and framework. The authors have shown rigor in their approach, collaborating with healthcare professionals and conducting comprehensive experiments.

The paper is generally well-structured and clearly written. The authors provide a good explanation of their methodology, including detailed descriptions of the dataset creation process and the CheckEHR framework stages. However, some technical aspects, particularly in the experimental results section, could benefit from further clarification.

The work presents several novel contributions, including the EHRCon dataset and the CheckEHR framework. The approach of using large language models for consistency checking in EHRs is innovative and represents a new direction in healthcare informatics. The potential impact of this work on improving healthcare documentation and reducing medical errors is substantial, making it highly significant for both the machine learning and healthcare communities.

Pros:

-	Addresses a crucial problem in healthcare informatics with real-world implications.
-	Novel dataset (EHRCon) that can serve as a benchmark for future research.
-	Innovative framework (CheckEHR) leveraging large language models for consistency checking.
-	Dual schema approach (MIMIC-III and OMOP CDM) enhances generalizability.
-	Comprehensive evaluation using multiple language models and experimental settings.
-	Collaboration with healthcare professionals ensures practical relevance.

Cons:

-	Overall performance of CheckEHR, while promising, still has room for improvement.
-	No clear guidance on handling detected inconsistencies in practice.
-	Assumption that inconsistencies originate from clinical notes rather than structured data.
-	Reliance on LLMs introduces risks of hallucination and inconsistency.

**Strengths:**

-	This work addresses an important issue in healthcare informatics - inconsistencies between clinical notes and structured data in Electronic Health Records (EHRs). This problem has direct implications for patient safety and care quality.
-	The authors provide a carefully curated dataset for consistency checking in EHRs – a significant contribution. It provides a valuable resource for future research in this area.
-	Both MIMIC-III and OMOP CDM schemas are used, which enhance the dataset's applicability and generalizability across different EHR systems.
-	The involvement of healthcare professionals in designing the labeling instructions adds credibility and ensures real-world relevance.
-	The paper includes thorough experiments using multiple large language models in both few-shot and zero-shot settings, providing a good understanding of the framework's capabilities and limitations.

**Additional Feedback:**

-	In Table 2 and Table 3, specify what the bolded numbers signify in the table captions.

**Documentation:**

I would recommend using another format other than pickle files to distribute the data. There can be issues for users (e.g., incompatible versions) when trying to load pickled files.

**Ethics:**

No ethical concerns that warrant further review.

**Limitations:**

-	The authors don’t discuss clear recommendations on how to handle or resolve detected inconsistencies in practice.

**Opportunities For Improvement:**

-	Confidence intervals for the metrics in Table 2 and Table 3 would help the reader understand whether the results statistically significantly differ or not.

**Relation To Prior Work:**

The authors do a nice job of clearly describing their work in the context of prior work.

**Summary And Contributions:**

This paper introduces EHRCon, a new dataset and task designed to check consistency between unstructured clinical notes and structured tables in Electronic Health Records (EHRs). The authors also present CheckEHR, a framework that leverages large language models to verify this consistency.
Contributions include:

1.	EHRCon Dataset:
-	Contains 3,943 manually annotated entities from 105 clinical notes, checked against database entries for consistency.
-	Available in two versions: one using the original MIMIC-III schema and another using the OMOP Common Data Model (CDM) schema, enhancing its applicability and generalizability.

2.	CheckEHR Framework:
-	An 8-stage process for verifying consistency between clinical notes and database tables using large language models.
-	Stages include note segmentation, named entity recognition, time filtering, table identification, pseudo table creation, self-correction, value reformatting, and query generation.

3.	Experimental Results:
-	Comprehensive evaluation using multiple large language models (Tulu2, Mixtral, Llama-3, GPT-3.5) in both few-shot and zero-shot settings.
-	Best performance achieved 61.06% recall in the few-shot setting using GPT-3.5.

---

> ### Author Rebuttal · Authors · 2024-08-17
>
> **Q4. Reliance on LLMs introduces risks of hallucination and inconsistency.**
>
> A4: Extracting entities related to EHR tables and their corresponding values from long clinical notes requires a high-level understanding of the note. To enable this setting with few-shot learning, we propose a framework that leverages the reasoning abilities of LLMs. As you noted, LLMs can present the risk of hallucination, and we have documented such issues (e.g., the creation of pseudo tables based on unit information not recorded in the notes) in Appendix J. To overcome the hallucination issue mentioned, we have carefully integrated the self-correction stage into our methodology, which has proven effective in addressing some of these challenges. However, we acknowledge that there is still room for improvement. We are committed to refining our approach and will diligently explore further solutions in our future research to minimize these risks even further.
>
>
> **Q5. Confidence intervals for the metrics in Table 2 and Table 3 would help the reader understand whether the results statistically significantly differ or not.**
>
> A5: We plan to conduct additional experiments to provide confidence intervals for the metrics. Although the limited time before the rebuttal deadline may prevent us from completing all experiments, we intend to report the results before the discussion period ends, if possible. Regardless, we will ensure that confidence intervals are included in the final version of the paper.
>
> **Q6. I would recommend using another format other than pickle files to distribute the data. There can be issues for users (e.g., incompatible versions) when trying to load pickled files.**
>
> A6: We have uploaded the pickle files to PhysioNet, and they are currently under review. To improve distribution, we will also include them in JSON format.
>
> **Q7. In Table 2 and Table 3, specify what the bolded numbers signify in the table captions.**
>
> A7:  Thank you for the valuable feedback. In Table 2, we have highlighted in bold the highest values for Recall, Precision, and Intersection under both the zero-shot setting and the few-shot setting. In Table 3, we have bolded the values to indicate which database, either MIMIC-OMOP or MIMIC-III, yields higher results for each model. We will incorporate this additional explanation into the final version.

---

> > ### Comment · Reviewer_V5Jt · 2024-08-21
> >
> > Thank you for a thorough response to my concerns, questions, and suggestions.
> >
> > To summarize, these paper updates I believe are needed:
> > - Q2/A2 - Guidance on handling detected inconsistencies in practice.
> > - Q5/A5 - Confidence intervals for metrics in Tables 2 & 3.
> > - Q6/A6 - Files uploaded in JSON format.
> > - Q7/A7 - Specify what the bolded numbers signify in the table captions.
> >
> > Other than these points, my concerns are alleviated. Nicely done.

---

> > > ### Author Rebuttal · Authors · 2024-08-21
> > >
> > > Thank you very much for your valuable feedback on our research. We truly appreciate your suggestions, which have greatly contributed to improving our work.
> > > In response to your recommendations, we will ensure that the final version includes additional guidance on handling inconsistencies, adds confidence intervals, and clarifies the significance of bold numbers. Additionally, we will provide the data in JSON format.
> > >
> > > Should you have any further questions or require additional clarification, please feel free to reach out at any time. We are committed to addressing all your concerns.
> > >
> > > Thank you once again for your thoughtful feedback.

---

> ### Author Rebuttal · Authors · 2024-08-17
>
> We are grateful for your valuable feedback and suggestions. We address your comments by following Q-A pairs (from **Q1** to **Q7**).
>
> **Q1. Overall performance of CheckEHR, while promising, still has room for improvement.**
>
> A1: Thank you for your feedback on the overall performance of CheckEHR. As you mentioned, achieving a recall performance of 61% may not yet be sufficient for real-world application, which indeed underscores the need for further enhancements.
> However, we would like to highlight that in our experiments, when the LLM performed consistency checks directly, it achieved a maximum recall of 10.12% (Please see the table below) . Given that our work tackles a novel task, this result represents a current state-of-the-art performance.
> We will prioritize improving this aspect in our future work. Our goal is to develop a framework that can be reliably and effectively deployed in practical scenarios. Your insights are invaluable to us, and we are determined to refine our approach further.
>
> -----
> | Model             | Recall  | Precision |
> |-------------------|---------|-----------|
> | ChatGPT 3.5 16k   | 10.12  | 19.88   |
> | Llama 3.1 70B     | 6.28   | 29.66    |
> | Mixtral 8X7B      | 0.00   | 0.00     |
>
>
>
>
>
>
> **Q2. No clear guidance on handling detected inconsistencies in practice.**
>
> A2: In actual practice, when practitioners write clinical notes, a real-time consistency check with tables can be performed. If any inconsistencies are detected, the practitioners can be notified immediately to correct them. We plan to include this guidance in the final version.
>
>
> **Q3. Assumption that inconsistencies originate from clinical notes rather than structured data.**
>
> A3: As discussed in Lines 97-104, we recognize that errors can occur in both clinical notes and structured data. Therefore, rather than assigning inconsistencies to a particular source, we have adopted a more flexible approach by categorizing labels as Consistent or Inconsistent, without making specific assumptions.

---

### Official Review · Reviewer_8FTT · 2024-07-25
**Interesting contribution; some limitations in validation**

**Rating:** 6
**Confidence:** 4
**Clarity:** The paper is very well written, and t…

**Review:**

This work introduces an important task and dataset to the field. However, there are some methodological shortcomings which significantly diminish its impact in its current state. Future work should address these limitations, particularly the validation issues noted below. Despite its limitations, this study lays a foundation for an important area of research.

Pros:

- Addresses an important problem in healthcare informatics with patient safety implications
- Introduces a novel dataset and task created in collaboration with healthcare professionals
- The two schema versions (MIMIC-III and OMOP CDM) increase generalizability
- Presents a multi-stage approach with the CheckEHR framework
- Offers a detailed annotation process and Item Search Tool for addressing clinical note-database discrepancies
- Includes a component analysis of the framework

Cons:

- Lack of validation for annotator-generated SQL queries
- No reporting of inter-annotator agreement
- Experimental design conflates model capabilities with framework effectiveness
- Fails to provide baselines using LLMs without the CheckEHR framework
- Does not include long-context models for processing entire clinical notes
- Limited dataset scope (single medical center, dated 2001-2012) and size (3,943 entities across 105 notes)
- Best model performance (61.06% recall) suggests limited success with CheckEHR framework

**Strengths:**

- The study introduces a novel dataset and task. It addresses an important problem in the field.
- The paper provides a thorough description of the annotation process.
- The CheckEHR framework is a novel approach to automating consistency checks.
- The authors conduct an informative ablation study to assess the contribution of different stages in their framework.
- The study addresses real-world issues in EHRs, such as abbreviations, common names, and brand names in clinical documentation.
- If further developed and validated, this work has clear implications for improving patient safety.

**Additional Feedback:**

I believe that the dataset and benchmark are likely to be more influential than CheckEHR, which seems more useful for weak LLMs.

**Correctness:**

The main potential issues with correctness come from the lack of annotation validation and limitations in the evaluation of CheckEHR. These issues are discussed in the Opportunities for Improvement section above.

**Documentation:**

The data appears to be well documented, though the files are currently only available as .pkl.

**Ethics:**

There are no ethical concerns.

**Limitations:**

The authors identify real limitations with their work at the end of the paper, which they are to be commended for. They do not address the limitations described above. I do not see any potentially negative social impacts of the work.

**Opportunities For Improvement:**

- The SQL queries generated by annotators were not validated. This could potentially lead to errors in consistency assessments.
- No inter-annotator agreement was reported, making it difficult to judge annotation reliability. Without this metric, it is not clear how reliable the annotations are.
- The experiments conflate model ability with the effectiveness of the CheckEHR framework. The study should have evaluated LLMs directly on the task without using CheckEHR. This makes it difficult to determine the actual value added by the framework.
- The study does not use long-context models, which could process the entire clinical note without losing potentially important context. Using such models could potentially eliminate the need for note segmentation.
- The dataset is relatively small, with only 3,943 entities across 105 clinical notes. This limited scale may not capture the full complexity and variety of real-world EHR inconsistencies.
- The best model achieves only 61% recall. This performance level suggests that significant improvements are needed before the system could be considered for real-world deployment. This is clearly not a problem for the benchmark part of the paper, but CheckEHR seems to be intended as an independent contribution.

**Relation To Prior Work:**

The discussion of related work is somewhat thin, though clearly space is limited. There is limited positioning of the work relative to other work on EHR analysis and benchmarking.

**Summary And Contributions:**

The paper introduces EHRCon, a dataset and task for checking consistency between clinical notes and structured tables in Electronic Health Records (EHRs). EHRCon contains manually annotated entities from MIMIC-III, available in both MIMIC-III and OMOP CDM schemas. The authors also present CheckEHR, an eight-stage process for eliciting consistency checks from LLMs. The study provides benchmark results for several models in zero-shot and few-shot settings, and a component analysis of the framework. The benchmarking performance suggests significant room for improvement in performing the task.

---

> ### Author Rebuttal · Authors · 2024-08-17
>
> **Q6. Best model performance (61.06% recall) suggests limited success with CheckEHR framework. The best model achieves only 61% recall. This performance level suggests that significant improvements are needed before the system could be considered for real-world deployment. This is clearly not a problem for the benchmark part of the paper, but CheckEHR seems to be intended as an independent contribution.**
>
> A6: The 61% recall performance indeed indicates that CheckEHR may face challenges in real-world deployment, as you have pointed out. However, it's important to note that without the use of our framework, directly applying LLM for consistency checks results in a recall performance of only 10.12%. Considering that this study addresses a novel task, our framework can still be regarded as achieving state-of-the-art results within this context.
>
>
> **Q7. The discussion of related work is somewhat thin, though clearly space is limited. There is limited positioning of the work relative to other work on EHR analysis and benchmarking.**
>
>
> A7: To the best of our knowledge, there has not been any prior research specifically focused on consistency checks between clinical notes and EHR tables. However, there have been studies that separately analyze EHR tables and databases.
> Numerous prior studies have explored table-based QA tasks [1,2,3,4,5]. For instance, Wang et al. [3] introduced the MIMICSQL dataset for a text-to-SQL generation task based on the MIMIC-III database. More recently, Lee et al. [4] developed EHRSQL, a new text-to-SQL dataset built on MIMIC-III and eICU.
>
> In terms of research involving clinical notes, Pampari et al. [6] presented emrQA, the first accessible patient-specific EMR QA dataset, consisting of 400,000 question-answer pairs and 1 million question-logical form pairs. Fan [7] generated why-questions based on discharge summaries, and Kweon et al. [8] developed the EHRNoteQA dataset focused on individual patients' discharge summaries. Additionally, research such as DrugEHRQA [9] has combined both structured tables and unstructured clinical notes of an EHR to answer questions. These studies are fundamentally different from ours as they focus on datasets specifically designed for QA tasks.
>
>
> We will ensure that the final manuscript provides a more comprehensive positioning of our work relative to these and other studies, highlighting the unique aspects of our approach.
>
> ---
> References:
> * [1] Seongsu Bae, Daeyoung Kim, Jiho Kim, and Edward Choi. Question answering for complex electronic health records database using unified encoder-decoder architecture. In Machine Learning for Health, pages 13–25. PMLR, 2021.
>
> * [2] Nicholas J. Dobbins, Bin Han, Weipeng Zhou, Kristine Lan, H. Nina Kim, Robert Harrington, Ozlem Uzuner, and Meliha Yetisgen. LeafAI: Query generator for clinical cohort discovery rivaling a human programmer. arXiv preprint arXiv:2304.06203, 2023.
>
> * [3] Ping Wang, Tian Shi, and Chandan K. Reddy. Text-to-SQL generation for question answering on electronic medical records. In Proceedings of The Web Conference 2020, pages 350–361, 2020.
>
> * [4] Gyubok Lee, Hyeonji Hwang, Seongsu Bae, Yeonsu Kwon, Woncheol Shin, Seongjun Yang, Minjoon Seo, Jong-Yeup Kim, and Edward Choi. EHRSQL: A practical text-to-SQL benchmark for electronic health records. Advances in Neural Information Processing Systems, 35:15589–15601, 2022.
>
> * [5] Sarvesh Soni, Surabhi Datta, and Kirk Roberts. QUEHRY: A question answering system to query electronic health records. Journal of the American Medical Informatics Association, 30(6):1091–1102, 2023.
>
> * [6] Anusri Pampari, Preethi Raghavan, Jennifer Liang, and Jian Peng. EMRQA: A large corpus for question answering on electronic medical records. arXiv preprint arXiv:1809.00732, 2018.
>
> * [7] Jungwei Fan. Annotating and characterizing clinical sentences with explicit why-QA cues. In Proceedings of the 2nd Clinical Natural Language Processing Workshop, 2019.
>
> * [8] Sunjun Kweon, Jiyoun Kim, Heeyoung Kwak, Dongchul Cha, Hangyul Yoon, Kwanghyun Kim, Seunghyun Won, and Edward Choi. EHRNoteQA: A patient-specific question answering benchmark for evaluating large language models in clinical settings. Preprint, arXiv:2402.16040, 2024.
>
> * [9] Bardhan, Jayetri, Anthony Colas, Kirk Roberts, and Daisy Zhe Wang. DRUGEHRQA: A question answering dataset on structured and unstructured electronic health records for medicine-related queries. arXiv preprint arXiv:2205.01290, 2022.

---

> ### Author Rebuttal · Authors · 2024-08-17
>
> **Q4. Does not include long-context models for processing entire clinical notes. The study does not use long-context models, which could process the entire clinical note without losing potentially important context. Using such models could potentially eliminate the need for note segmentation.**
>
> A4: Thank you for your valuable suggestion regarding the use of long-context models for processing entire clinical notes. Based on your feedback, we conducted a thorough evaluation of CheckEHR without the note segmentation stage, testing on 25% of our total test set (Please see the table below). We utilized long-context models such as ChatGPT-3.5 16k, LLaMA 3.1 70B, and Mixtral 8X7B. While we observed some improvement in precision in certain cases, there was also a consistent decrease in recall across all scenarios. This indicates that current LLMs may face challenges in accurately extracting detailed information from free-form long contexts. Since our primary goal is error detection, including the entire long context may not always lead to the best results. Moreover, our note segmentation method allows us to use models with shorter context lengths, potentially making our approach more applicable across different hospitals.
>
> Due to time and budget constraints during the rebuttal period, we were able to experiment with only 25% of the test set. However, we are committed to including results from the entire test set in the final version to provide a more comprehensive understanding. We appreciate your consideration and hope that this additional analysis will further demonstrate the robustness of our approach.
>
> -----
> | Model                                | Recall | Precision |
> |--------------------------------------|--------|-----------|
> | ChatGPT 3.5 4k (note segmentation)    | 65.45  | 50.75     |
> | ChatGPT 3.5 16k                      | 48.64  | 54.53     |
> | Llama 3 70B (note segmentation)       | 53.86  | 47.01     |
> | Llama 3.1 70B                        | 35.23  | 45.40     |
> | Mixtral 8X7B (note segmentation)      | 54.92  | 44.60     |
> | Mixtral 8X7B                          | 36.43  | 42.58     |
>
>
>
> -----
> **Q5. Limited dataset scope (single medical center, dated 2001-2012) and size (3,943 entities across 105 notes). The dataset is relatively small, with only 3,943 entities across 105 clinical notes. This limited scale may not capture the full complexity and variety of real-world EHR inconsistencies.**
>
> A5: We understand the concerns regarding the use of data from a single medical center and the relatively small size of the dataset. However, given that MIMIC is the only publicly available EHR database offering paired notes and tables, this dataset was the most appropriate and available choice for our study. Regarding the size of the dataset, as we discussed in the limitations section of our paper, we fully recognize this as a limitation of our work. Nevertheless, our primary goal was to define a new task, and to this end, we have focused on creating a high-quality dataset, even with the limited amount of human annotation. We sincerely hope that by leveraging more scalable methods in the future, we will be able to expand the scope of our research further.

---

> ### Author Rebuttal · Authors · 2024-08-17
>
> We are grateful for your valuable feedback and suggestions. We address your comments by following Q-A pairs (from **Q1** to **Q7**):
>
> **Q1. Lack of validation for annotator-generated SQL queries. The SQL queries generated by annotators were not validated. This could potentially lead to errors in consistency assessments.**
>
> A1: Thank you for your valuable feedback regarding the lack of validation for the SQL generated by the annotators. We understand the importance of ensuring the reliability of these SQL queries, and we would like to clarify our approach.
> The SQL queries were generated using a template-based method during our data annotation process. Annotators filled in specific slots within predefined SQL templates to create the necessary queries. For example, a typical template might look like this: `SELECT * FROM {table1} JOIN {table2} ON {table1}.{column_name1} = {table2}.{column_name2} WHERE {table1}.hadm_id = {hadm_id} AND [time_value_template] AND {table1}.[condition_value_column] = {condition_value}`
> The [time_value_template] includes conditions like: `strftime(‘%Y-%m-%d’, {table}.{time_column}) BETWEEN strftime(‘%Y-%m-%d’, datetime(‘{admission}‘, ‘-1 day’)) AND strftime(‘%Y-%m-%d’, datetime(‘{admission}‘, ‘+1 day’))`
>
> To ensure the accuracy of these templates, two of the authors meticulously validated each one. To confirm the reliability of the templates, they reviewed all possible queries in addition to the query results. In total, we used 27 different templates, each of which underwent rigorous validation checks.
> We believe this template-based approach, combined with careful validation, provides a high level of reliability for the generated SQL queries. We will ensure that all relevant details about the templates and the validation process are thoroughly documented in the final paper.
>
>
>
> **Q2. No reporting of inter-annotator agreement. No inter-annotator agreement was reported, making it difficult to judge annotation reliability. Without this metric, it is not clear how reliable the annotations are.**
>
> A2: As detailed in Appendix E, we implemented a thorough quality control process to ensure the reliability of our annotations. After completing all labeling tasks, the annotators conducted cross-validation on a random sample of 20 notes, resulting in an F1 score of 0.880 for entity recognition. Additionally, in instances where both annotators identified the same entities, the labels were consistent 93.8% of the time. These results demonstrate a high level of agreement between the annotators, underscoring the reliability of our annotations.
>
>
>
>
>
> **Q3. Experimental design conflates model capabilities with framework effectiveness. The experiments conflate model ability with the effectiveness of the CheckEHR framework. The study should have evaluated LLMs directly on the task without using CheckEHR. This makes it difficult to determine the actual value added by the framework.**
>
> A3: Thank you very much for your insightful feedback. We greatly appreciate your concern regarding the potential conflation of model capabilities with the effectiveness of the CheckEHR framework. In response, we conducted an additional evaluation to determine whether an LLM could independently verify consistency between unstructured clinical notes and structured tables without relying on the CheckEHR framework.
> For this experiment, we provided the LLM with all the information initially supplied to CheckEHR (such as the definition of time, NER scope, table and column descriptions, etc.) and tasked it with directly performing the consistency check between the clinical notes and EHR tables. The results for 25% of the test set, as shown in the table below, indicate that attempting this task without a carefully designed framework like CheckEHR leads to significantly lower performance (achieved a maximum recall of 10.12), highlighting the complexity and challenges inherent in this task.
>
> Although we were limited by time and resources and could only conduct this experiment on 25% of the test set, we are fully committed to extending this evaluation to cover the entire test set. We assure you that the final version will include results from the full test set to provide a comprehensive comparison.
>
> -------
> | Model             | Recall  | Precision |
> |-------------------|---------|-----------|
> | ChatGPT 3.5 16k   | 10.12  | 19.88    |
> | Llama 3.1 70B     | 6.28   | 29.66    |
> | Mixtral 8X7B      | 0.00   | 0.00     |

---

> ### Author Response · Authors · 2024-08-25
> **Sincerely expecting further discussions with Reviewer 8FTT**
>
> Dear Reviewer 8FTT,
>
> We sincerely appreciate your thorough review of our paper and the invaluable insights you have provided. As the discussion period is drawing to a close, we would like to inquire if our response addressed your concerns.
> Please let us know if you have any further questions or feedback.

---

> > ### Comment · Reviewer_8FTT · 2024-08-26
> >
> > Thank you for your responses. I have increased my score.
> >
> > > To ensure the accuracy of these templates, two of the authors meticulously validated each one. To confirm the reliability of the templates, they reviewed all possible queries in addition to the query results. In total, we used 27 different templates, each of which underwent rigorous validation checks. We believe this template-based approach, combined with careful validation, provides a high level of reliability for the generated SQL queries. We will ensure that all relevant details about the templates and the validation process are thoroughly documented in the final paper.
> >
> > Could you say more about the validation for the queries? What do you mean that you reviewed all possible queries in addition to the query results?
> >
> > Re: benchmarking of long context models and CheckEHR. None of the strongest LLMs were evaluated on the task, making the benchmarking results difficult to interpret for the reasons described in my review.

---

> > > ### Author Rebuttal · Authors · 2024-08-26
> > >
> > > Thank you very much for your valuable feedback on our research. We truly appreciate your suggestions, which have greatly contributed to improving our work. We address your comments by following Q-A pairs.
> > >
> > > **Q1. Could you say more about the validation for the queries? What do you mean that you reviewed all possible queries in addition to the query results?**
> > >
> > > A1: We reviewed a total of 9 tables (*i.e.*, chartevents, labevents, diagnoses_icd, procedures_icd, prescriptions, outputevents, inputevents_mv, inputevents_cv, and microbiologyevents) in combination with different time formats (please refer to section 3.1 Time Expression and Appendix D). This resulted in a total of 27 query templates, as detailed in the attached PDF file.
> > >
> > >
> > > The order of the query templates in the table follows this sequence: event time written in a standard time format, event time described in a narrative style, and events where time information is not explicitly stated. If the time is expressed with HH-MM-SS, it should be written as follows: `strftime('%Y-%m-%d %H:%M:%S', {table}.{timecolumn})`. Examples of the narrative style include terms like "admission," "discharge," and "yesterday". In cases where time information is not provided, for nursing and physician notes, we check whether a clinical event occurred within one day before or after the charted date (charttime).
> > >
> > > Through manual review of all notes, we discovered that for the tables chartevents, labevents, inputevents_mv, and prescriptions, there are instances where dates need to be calculated, such as when the notes reference time in phrases like "hospital day 3". In these cases, we calculate the event time based on the admission date.  For the case of procedure_icd and diagnoses_icd, there is only one template available as these tables are used for insurance claims and do not contain any time-related columns.
> > >
> > > These templates were tested by two authors. They filled in the slots to create SQL queries and manually checked the query results against the database to ensure the templates were correct. The annotators then used these templates for annotation, helping to maintain accuracy. The results of these annotations were cross-checked (see Appendix E), and a 93.8% agreement rate was observed, which further indirectly validates the correctness of the SQL queries generated using these templates.
> > >
> > > ----------
> > > **Q2. benchmarking of long context models and CheckEHR. None of the strongest LLMs were evaluated on the task, making the benchmarking results difficult to interpret for the reasons described in my review.**
> > >
> > > A2: Due to privacy protection requirements for the MIMIC dataset, it was necessary to use HIPAA-compliant models provided by Azure. As a result, we were limited to conducting experiments with API-based models offered by Azure, specifically the ChatGPT-based models. This restriction prevented us from experimenting with models such as Gemini and Claude.
> > >
> > > Furthermore, the cost of models like GPT-4 is approximately thirty times higher than that of ChatGPT-3.5, making it impractical to use GPT-4 for extensive experiments due to budget constraints. To address your concerns as much as possible within these limitations, we conducted our experiments using ChatGPT-3.5 16k, Llama 3.1, and Mixtral.

---

### Author Rebuttal · Authors · 2024-08-19

The EHRCon dataset has been accepted by PhysioNet, and the citation information has been updated. After final copyediting and approval, it will soon be publicly available.

The updated citation information is as follows:
*Kwon, Y., Kim, J., Lee, G., Bae, S., Kyung, D., Cha, W., Pollard, T., Johnson, A., & Choi, E. (2024). EHRCon: Dataset for Checking Consistency between Unstructured Notes and Structured Tables in Electronic Health Records (version 1.0.0). PhysioNet. https://doi.org/10.13026/x1ea-np24*

(Note that the DOI will not be activated until the dataset is published).

---

### Decision · Program_Chairs · 2024-09-26

**Decision:**

Accept (Spotlight)

**Comment:**

The paper introduces EHRCon, a novel dataset for verifying the consistency between unstructured clinical notes and structured Electronic Health Record (EHR) tables, and presents CheckEHR, an eight-stage framework that utilizes large language models (LLMs) to automate this consistency checking. The study evaluates several LLMs in few-shot and zero-shot settings, achieving a maximum recall of 61.06%. The paper addresses an important issue in healthcare informatics and makes a significant contribution by offering a benchmark dataset and framework that can be used for future research.

Strengths:
1) Novel Contribution: The introduction of EHRCon fills a gap in healthcare informatics, providing a benchmark dataset and a new task to ensure data consistency between unstructured and structured EHR data.
2) CheckEHR Framework: The structured eight-stage approach to verifying consistency is innovative and demonstrates potential for automating error detection in clinical documentation.
3) Comprehensive Evaluation: The study rigorously tests the framework across multiple LLMs and both few-shot and zero-shot settings, providing detailed insights into the framework’s performance.
4) Dataset Applicability: The use of both MIMIC-III and OMOP CDM schemas enhances the dataset’s generalizability across different EHR systems, making it relevant for broader use.
5) Collaboration with Healthcare Professionals: Involving clinicians in the dataset creation process ensures that the annotations and task align with real-world healthcare challenges.

Opportunities For Improvement:
1) SQL Query Validation: The study did not initially validate SQL queries generated by annotators, raising concerns about the reliability of the consistency checks. The authors later clarified their validation method, but this remains a notable gap.
2) Inter-Annotator Agreement: While the authors later provided a 93.8% inter-annotator agreement, this was not initially reported and is important for establishing the reliability of the annotations.
3) Limited Dataset Size: The dataset is relatively small, with only 3,943 entities across 105 clinical notes, and comes from a single medical center. This limits the generalizability of the findings.
4) Model Performance: The best model achieved only 61% recall, which, while an improvement over baseline performance, indicates that the system requires further development before it can be applied in practice.
5) Evaluation Design: The evaluation of LLMs was conducted within the CheckEHR framework, without direct comparison to baseline LLM performance outside the framework, which makes it difficult to isolate the framework’s contribution.
6) Handling Detected Inconsistencies: The paper lacks clear guidance on how detected inconsistencies should be addressed in practical applications.